# PD-L1$^+$ and XCR1$^+$ dendritic cells are region-specific regulators of gut homeostasis

Thais G. Moreira [1]✉, Davide Mangani [1], Laura M. Cox [1], Jeffrey Leibowitz[1], Eduardo. L. C. Lobo [1], Mariana A. Oliveira[2], Christian D. Gauthier[1], Brenda N. Nakagaki[1], Valerie Willocq[1], Anya Song [1], Lydia Guo [1], David C. A. Lima[2], Gopal Murugaiyan [1], Oleg Butovsky [1], Galina Gabriely[1], Ana C. Anderson [1], Rafael M. Rezende[1], Ana Maria C. Faria[2] & Howard L. Weiner [1]

The intestinal mucosa constitutes an environment of closely regulated immune cells. Dendritic cells (DC) interact with the gut microbiome and antigens and are important in maintaining gut homeostasis. Here, we investigate DC transcriptome, phenotype and function in five anatomical locations of the gut lamina propria (LP) which constitute different antigenic environments. We show that DC from distinct gut LP compartments induce distinct T cell differentiation and cytokine secretion. We also find that PD-L1$^+$ DC in the duodenal LP and XCR1$^+$ DC in the colonic LP comprise distinct tolerogenic DC subsets that are crucial for gut homeostasis. Mice lacking PD-L1$^+$ and XCR1$^+$ DC have a proinflammatory gut milieu associated with an increase in Th1/Th17 cells and a decrease in Treg cells and have exacerbated disease in the models of 5-FU-induced mucositis and DSS-induced colitis. Our findings identify PD-L1$^+$ and XCR1$^+$ DC as region-specific physiologic regulators of intestinal homeostasis.

[1] Evergrande Center for Immunologic Diseases and Ann Romney Center for Neurologic Diseases, Department of Neurology, Brigham and Women's Hospital, Harvard Medical School, Boston, MA, USA. [2] Departamento de Bioquímica e Imunologia, Instituto de Ciências Biológicas, Universidade Federal de Minas Gerais, Belo Horizonte, MG, Brazil. ✉email: tmoreira@bwh.Harvard.edu

A beneficial multi-ecosystem involving commensal micro-organisms, dietary antigens, and the gut-associated lymphoid cells is orchestrated by several immune cell types that promote tolerance in the intestine. It is known that many types of immune cells are required for the maintenance of intestinal homeostasis[1]. Dendritic cells (DC) play a major role because of their ability to bridge innate and adaptive immune responses[2]. DCs are derived from DC precursors produced in the bone marrow that migrate to the blood and seed multiple tissues where they develop into two distinct lineages of conventional DCs (cDCs) based on their ontogeny: cDC1, which express the transcription factors IRF8, BATF3, and ID2[3] and cDC2 that express IRF4 and ZEB2[4]. In the intestinal mucosa, DCs are located diffusely throughout the intestinal lamina propria (LP), within gut-associated lymphoid tissues, including Peyer's patches (PP), lymphoid aggregates, and in intestinal draining lymph nodes[5,6] Tolerogenic DCs migrate to the draining lymph nodes where they promote tolerance by inducing regulatory T (Treg) cells[7,8]. However, the role of DCs in immune events that occur in the intestinal LP required for homeostatic responses to commensal microbes in the large intestine and dietary antigens in the upper Small Intestine (uSI) is not well understood.

cDC1s and cDC2s in the gut-draining lymph nodes consist of $CD103^+CD11b^-XCR1^+$ and $CD103^+CD11b^+Sirp\alpha^+$ DCs, respectively, with cDC1s being the primary inducers of Treg cells and oral tolerance[9]. Although DCs have been characterized in gut lymph nodes the features of intestinal LP are unknown. This is important as intestinal DCs have a key physiologic function in maintaining gut homeostasis. It is not clear whether DCs residing in distinct intestinal compartments are functionally unique and thus drive different T-cell responses.

Here, we show that DCs from different compartments of the gut promote distinct antigen-specific T-cell priming in vitro and in vivo. Using RNA sequencing and mass and flow cytometry, we characterize DCs throughout the intestine and show that the small intestine is enriched in $PD-L1^+$ DCs, whereas the large intestine is enriched in $XCR1^+$ DCs. Mice lacking programmed death 1 (PD-1)-$L1^+$ or $XCR1^+$ DCs exhibit a pro-inflammatory gut milieu associated with an increase in Th1/Th17 cells and a decrease in Treg cells in the models of 5-FU-induced mucositis and dextran sodium sulfate (DSS)-induced colitis. In humans, we show that PD-L1 is increased in DCs from the upper gut as compared with colonic biopsies from the same healthy individuals. Our findings identify regional-specific compartmentalization of distinct intestinal DC signatures at the level of adaptive immunity and outline a framework for understanding the refinement and robustness of gut homeostasis.

## Results

**Immune cells are differentially distributed in specific anatomic regions of the gut**. The small and large intestines have unique anatomical and physiological characteristics. Food absorption takes place in the upper regions of the gut and terminal ileum, whereas the large intestine houses most of the vast diversity and quantity of commensal microbes that inhabit the gut[10]. Prior to a detailed investigation of gut immune cells, we first characterized the microbiota from different regions of the gut to establish that the microbiome from animals in our facility was not significantly different from what has been described[11–13]. We found two main microbial clusters, one associated with the small intestine (duodenum, jejunum, and ileum) and the other with the large intestine (cecum and colon) (Supplementary Fig. 1a). Microbiota α-diversity analysis revealed that the cecum microbiota had the highest phylogenetic diversity (Supplementary Fig. 1b). *Firmicutes*, *Verrucomicrobia,* and *Bacteroidetes* were the most

abundant phyla in the mouse intestine and proteobacteria were the least abundant (Supplementary Fig. 1c). At the class level, the small intestine was largely populated by facultative anaerobes such as *Bacilli* that gradually decreased throughout the gut, whereas the ileal and large intestinal compartments were enriched with strict anaerobes such as *Clostridia* (Supplementary Fig. 1c). *Verrucomicrobiae* was dominant in the large intestine and less abundant in the small intestine (Supplementary Fig. 1c). At the species level, *Lactobacillus* was the most prevalent species followed by *Akkermansia*, representing 29.5 and 13.5% of all bacteria, respectively (Supplementary Data 1). The major species belonging to the Firmicutes and Verrucomicrobia phyla were oppositely distributed in the gut (Supplementary Fig. 1d). We used linear discriminant analysis effect size (LEfSe) to identify region-specific changes. The cladogram (Supplementary Fig. 1e) shows microbial taxa that were increased in the small intestine (green labels) and large intestine (red labels) from the kingdom to the genus phylogenetic levels. *Lactobacillus* and *Faecalibaculum rodentium* were increased in the small intestine whereas *Corynebacterium, Bacteroides, Clostridium, Akkermansia muciniphila, Parasutterella, Molilicutes,* and several undefined *Lachnospiraceae* and *Ruminococcaceae* were increased in the large intestine (Supplementary Data 2).

We then investigated immune cell populations in the LP from different regions of the gut using a mass cytometry (CyTOF) panel of 31 markers (Supplementary Data 3). We gated on CD45+ cells and found that markers associated with antigen-presenting cells including CD24, CD11b, CD11c, and F4/80 as well as CD135, PD-L1, CD39, Sirpα, CCR7, and CD205 decreased from upper to lower regions of the intestine with the highest numbers in the duodenum and lowest numbers in the colon (Fig. 1a; Supplementary Fig. 2a). We also found an increase in B cells and $Foxp3^+$ Treg cells in the colon vs duodenum (Supplementary Fig. 1b). No changes in total $CD4^+$ and $CD8^+$ T cells were observed (Supplementary Fig. 2b). As shown in Fig. 1b, the frequency of $CD11b^+$ and $CD11c^+$ cells were significantly decreased in the ileum, cecum, and colon (range 2.0–3.3%) vs the duodenum and jejunum (range 25.2–33.5%). Of note, gut region-specific distribution and abundance of CD11c, CD11b, and different DC subsets were reported by Denning and colleagues[14]. Moreover, we found that 58.93% of live CD45+ cells in duodenum expressed CD24 vs 23.53% in the colon (Supplementary Fig 2a). We then generated a t-SNE plot to compare duodenum and colon and found five DC clusters in the duodenal LP that were not observed in the colonic LP (Fig. 1c). These cell clusters were defined by the expression of markers that identify DC subsets, including CD103, CX3CR1, CCR7, and Clec9a. We also investigated immune cells in the stomach and found that $CD11b^+$ and $CD24^+$ cells comprised >50% of CD45+ cells with >10% of the cells being $CD11c^+$ (Supplementary 1c, d). We further investigated $CD11c^+$ cells in the duodenum vs colon by immunofluorescence and immunohistochemistry and found an increased number of $CD11c^+$ cells (Fig. 1d, e). To confirm that $CD11c^+$ cells were enriched in the duodenum, counting beads were employed to quantify the absolute number of LP DCs. We found that the duodenum contained four times more $CD11c^+$ DCs ($F4/80^-CD64^-$) than the colon (Fig. 1f). The increased frequency and diversity of DCs in the duodenum vs colon suggests a differential immune response of the gut to food antigens present in the small intestine vs microbes in the large intestine.

**Unique DC signatures are found in specific regions of the gut**. We investigated DC gene expression, phenotype, and function in five intestinal compartments (duodenum, jejunum, ileum, cecum,

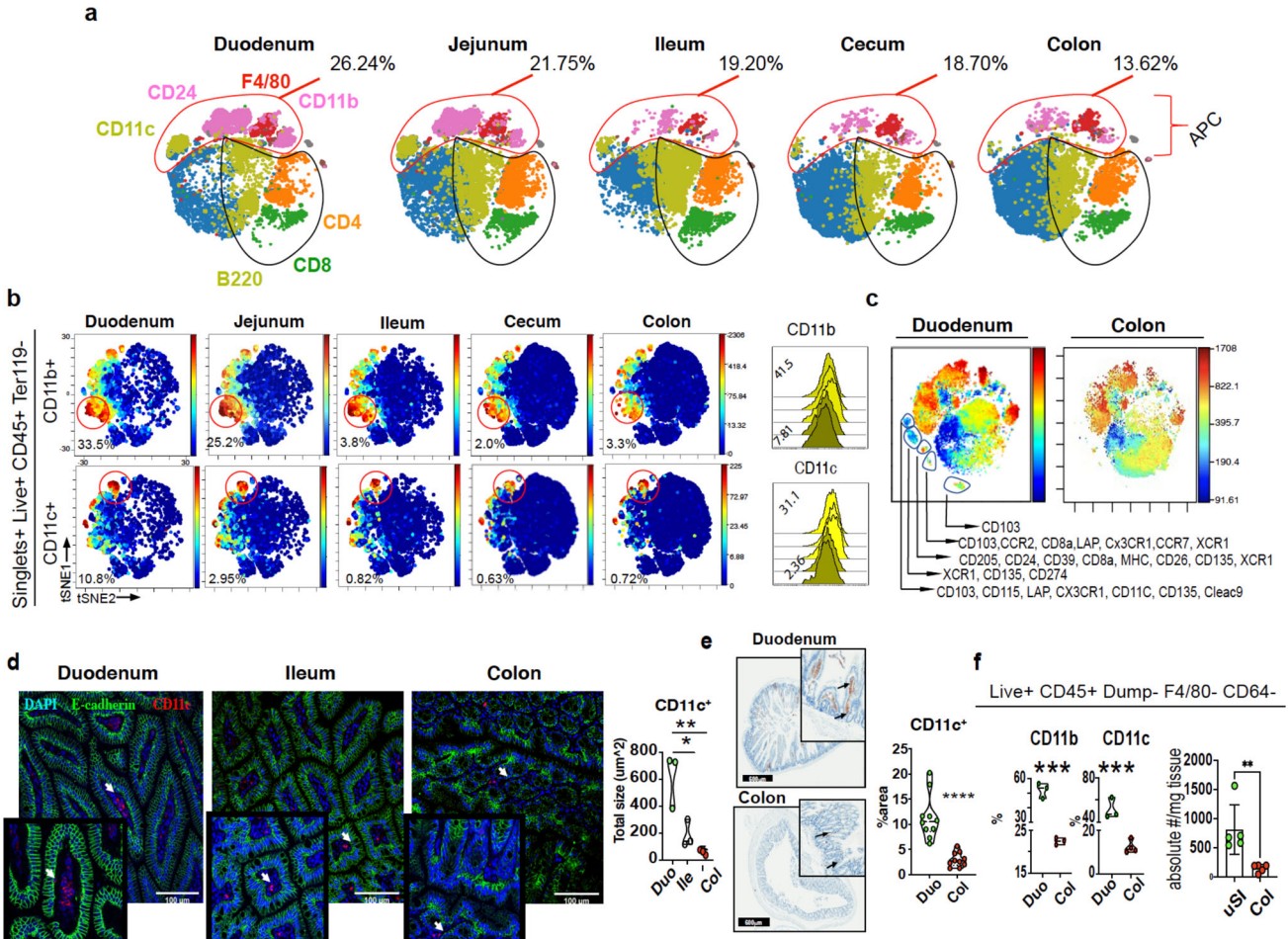

**Fig. 1 Immune cells are differentially distributed in specific gut regions. a** Mass cytometry t-SNE analysis in the LP of each intestinal region. **b** Mass cytometry t-SNE plots of different regions of gut LP. CD11b and CD11c clusters of CD45$^+$ gated cells are shown. **c** t-SNE plots comparing duodenal and colonic LP cells. All t-SNE plots correspond to concatenated data of singlets, live, CD45$^+$, and Ter119$^-$ gated cells. $N = 10$ mice/region of the gut over three independent experiments. **d** Immunofluorescence of duodenum, ileum, and colon. DAPI (blue) e-cadherin (green) and CD11c (red) (×40), the total size of fluorescently labeled CD11c from each sample was obtained using an ImageJ custom macro code. **e** Immunohistochemistry of CD11c$^+$ cells in duodenum and colon (×10). Percentage of area showing marked CD11c$^+$ DCs in the LP considering the entire tissue area. **f** Quantification of DCs using flow cytometry (live, CD45$^+$F4/80/CD64$^-$CD11c$^+$) per mg of tissue and counting beads. $N = 5$ over two independent experiments. Bars represent mean ± SEM. One-way ANOVA followed by Tukey post hoc test for multiple comparisons was used for IF comparison. Unpaired two-tailed Student's $t$ test was used for IHC and flow cytometry. *$p = 0.02$, **$p = 0.008$, ***$p = 0.0007$, ***$p = 0.005$ (CD11c),****$p < 0.0001$. Antigen presentation cells (APC).

and colon) by high throughput RNA sequencing (Smart-Seq2). We found differential DC gene expression in the five regions of the gut, though some genes were shared by DCs throughout the intestine (Fig. 2a) as shown by PCoA (Fig. 2b), heatmap (Fig. 2c), and volcano plots (Fig. 2d). *Cd274* (*Pd-l1*) and *Cd209b* were upregulated in the small intestine whereas *Xcr1* and *S100a4* were upregulated in the large intestine (Fig. 2e). Thus, DCs from different intestinal compartments have a unique gene signature.

To characterize intestinal DCs by protein expression, we performed CyTOF containing a panel of 38-metal conjugated DC markers (Supplementary Data 4; Fig. 2f, g; Supplementary Fig 2e–g for gating strategies) that we found by RNA-seq analysis. Consistent with our RNA-seq data, CyTOF analysis demonstrated unique DC subsets throughout the intestinal compartments (Fig. 2f). CyTOF and flow cytometry showed that CD101, PD-L1, CD209b, and Sirpα were upregulated in small intestinal DCs, whereas XCR1, TLR3, ICOSL, and S100a4 were upregulated in large intestinal DCs (Fig. 2f). Importantly, we found that PD-L1$^+$ DCs were enriched in the upper gut (84.1% in duodenum vs 22% in the colon), whereas XCR1$^+$ DCs were

enriched in the colon (12% vs 30.3%, respectively) (Fig. 2f). cDC2s (Sirpα$^+$) represented the most dominant DC subtype in all intestinal compartments, ranging from 85% of the DCs in the duodenum vs 69% in the colon. cDC2s located in the small intestine highly expressed PD-L1 (~86%) in duodenal DCs as compared with colonic DCs, which had significantly less PD-L1 expression (~23%) (Fig. 2g). Our results demonstrate that distinct cDC2s exist in different intestinal compartments and suggest that PDL1$^+$ duodenal DCs play a critical role in small intestinal tolerance and XCR1$^+$ DCs are important for tolerance in the large intestine.

**DCs from gut compartments induce distinct T-cell differentiation.** DCs are critical for adaptive immune responses owing to their unique role as antigen-presenting cells[15]. The role of regional-specific intestinal DCs in priming T cells has been proposed[14,16,17]. Because we found unique subsets of DCs differentially distributed throughout the gut, we investigated whether DCs from different intestinal compartments induced distinct

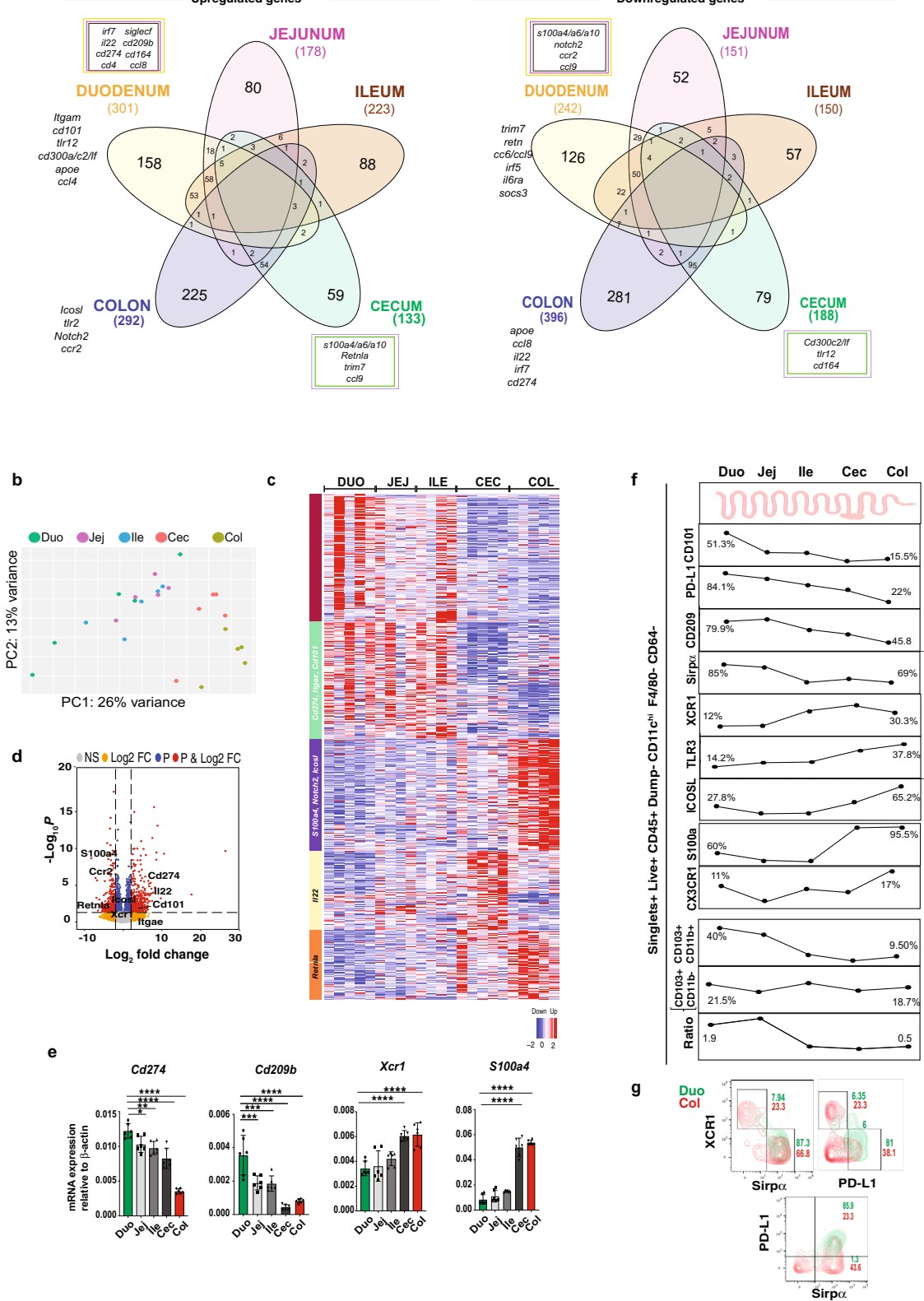

T-cell responses. We cultured DCs from the uSI (duodenum and proximal jejunum) or the distal colon with OT-II naive T cells (1:20 ratio) in the presence of ovalbumin (OVA) for 7 days and measured cytokine production and T-cell differentiation by flow cytometry. We found increased production of IL-13 and TNF in supernatants from cultures with uSI DCs and higher levels of IL-

22, IL-17A, IL-17F, IFN-γ, and IL-10 from cultures with colonic DCs. No difference between uSI vs colon was observed for IL-4 and IL-6 (Fig. 3a). To investigate whether cytokine production was associated with T-cell differentiation, we sorted post-cultured T cells and performed qPCR analysis of *Ifng*, *Tnf*, *Il6*, and *Il10*. We found that *Ifng* mRNA was upregulated in re-sorted T cells

**Fig. 2 DC transcriptome and markers showing tissue DC regional specialization. a** Diagram showing upregulation and downregulation of genes found in RNA-seq analysis from DCs sorted from duodenal, jejunal, ileal, cecal, and colonic LP of 8 to 10-week-old naive C57BL/6 J mice. **b** PCoA, **c** heatmap, and **d** volcano plots of genes found in RNA-seq analysis. $N = 5$. **e** RT-qPCR from sorted DCs of different regions of the gut showing *Cd274, Cd209, Xcr1,* and *S100a4* expression. ($N = 6$). Bars represent mean ± SEM. One-way ANOVA followed by Tukey post hoc test for multiple comparisons. *$p = 0.02$, **$p = 0.003$, Duo vs Jej ***$p = 0.006$ Duo vs Ile ***$p = 0.004$; ****$p < 0.0001$. **f** Mass and flow cytometry data showing the percentage of CD101, PD-L1, CD209b, Sirpα, XCR1, TLR3, ICOSL, CX3CR1 expression on DCs and CD103+CD11b− and CD103+CD11b+ cells gated on singlets, live, CD45+ CD19−B220− Nk1·1−Ly−6G−CD3− F4/80−CD64−CD11c^High. **g** Dot plots showing the percentage of PD-L1+, XCR1+, and Sirpα+ DCs LP from duodenum and colon by mass cytometry. $N = 5$; a pool of three mice/region of the gut over at least three independent experiments. Source data are provided as a Source Data file. Duodenum (DUO), jejunum (JEJ), ileum (ILE), cecum (Cec) colon (COL).

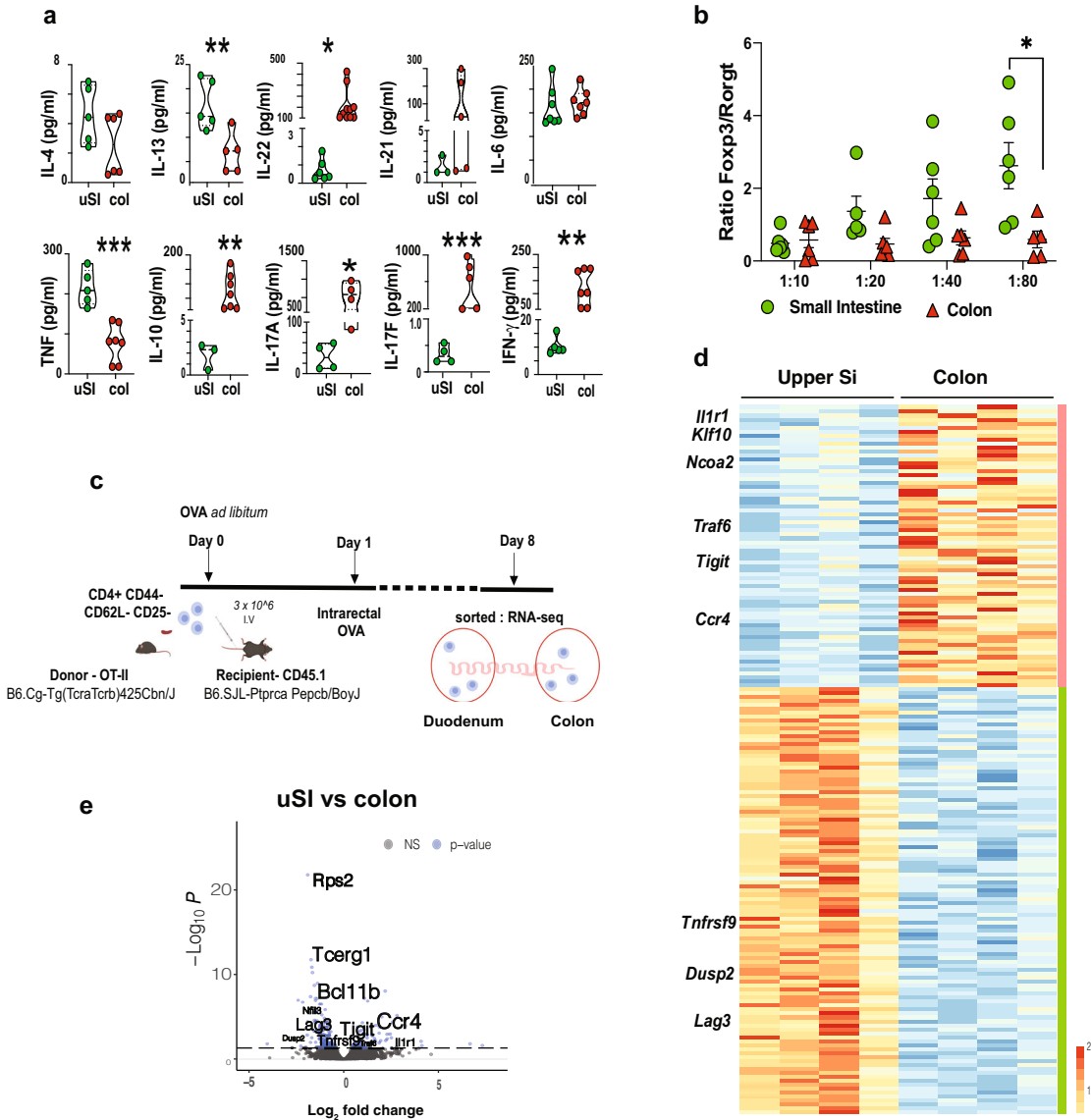

**Fig. 3 Gut regional DC induce distinct T-cell differentiation. a** Cytokine quantification by Legendplex from supernatants of upper small intestine (uSI) (duodenum and proximal jejunum) LP sorted DCs cultured with naive OT-II CD4+ T for 7 days in the presence of OVA peptide323-339. Unpaired two-tailed Student's *t* test. Boxes and bars represent mean ± SEM. *$p = 0.04$ (IL-22) *$p = 0.02$ (IL-17A), **$p = 0.009$ (IL-13), **$p = 0.01$ (IL-10), **$p = 0.004$ (IFN-γ), ***$p = 0.0002$ (TNF) ***$p = 0.001$ (IL-17F). **b** Foxp3 and RORγt ratios in different DC:T-cell ratios. Two-way ANOVA followed by Tukey post hoc test for multiple comparisons.*$p = 0.017$. $N = 5$; a pool of three mice/region of the gut over at least two independent experiments. **c** Adoptive transfer scheme. CD45.1 mice received $3 \times 10^6$ naive T cell (CD4+CD62L+CD44−CD25−) from OT-II mice. Mice were then given OVA (8 g/L) in the drinking water for 7 days and received an intra-colonic injection of OVA (2.4 mg/mice) 24 h after T-cell transfer. LP cells from the uSI and colon were then collected, and OT-II cells were sorted and sequenced by Smart–Seq2. **d** Heatmap and highlighted genes from OT-II cells sorted as described in **c**. **e** Volcano plot showing upregulated and downregulated genes in OT-II cells from **c**. Source data are provided as a Source Data file upper small intestine (uSI) and colon (col). Biorender was used for illustration on 3c.

that were co-cultured with colonic DCs and that *Tnf* mRNA was upregulated in T cells co-cultured with uSI DC. *Il6* mRNA was not detected in post-cultured T cells and *Il10* expression was very low at day 7 (Supplementary Fig. 3a). T-cell differentiation assessed by flow cytometry showed that an increase in DC: T-cell ratio was associated with an increase in T-cell differentiation (Fig. 3b). Moreover, an increased Treg/Th17 ratio was observed in the uSI vs colon (Fig. 3b; Supplementary Fig. 3b). We did not observe differences in T-cell proliferation related to DC gut location or DC:T-cell ratios in vitro (Supplementary Fig. 3c).

To investigate T-cell differentiation by DCs from different intestinal compartments in vivo, we utilized an adoptive T-cell transfer model in which CD45.1 C57BL/6 J congenic mice received $3 \times 10^6$ naive T cells (CD4+CD62L+CD44−Foxp3−) from CD45.2 OT-II mice and then were given OVA for 7 days in the drinking water (Fig. 3c). To ensure that OVA also reached the colon, mice received an intra-colonic injection of OVA 24 h after T-cell transfer (Supplementary movie 1, Supplementary Fig. 3d, e). T cells were then sorted from duodenal and colonic LP (Supplementary Fig. 3f–h) and sequenced by Smart-Seq2. We found that 205 genes were differentially expressed between OT-II cells harvested from the duodenum vs colon; 134 genes were upregulated in the duodenum and 71 genes upregulated in the colon (Fig. 3d, e). OT-II cells isolated from the duodenum and jejunum upregulated the activation markers *Tnfrsf9* (CD137) and *Lag3* (Fig. 3d, e), which have been associated with a T-cell regulatory phenotype[18,19]. We also found that *Bcl11b*, a nuclear factor implicated in both Th2 differentiation and Treg programing[20,21] was upregulated in OT-II T cells from the duodenal LP (Fig. 3e). OT-II cells sorted from the colonic LP also displayed a Treg cell signature, though different molecular pathways were observed (Fig. 3d, e) including *Tigit*, *Ccr4*, *Klf10*, and *Traf6*, which are implicated in downstream regulation of receptor families with immunoregulatory function[18,22–25]. As opposed to the upper gut, OT-II cells isolated from the colon had increased expression of genes involved in Th17 differentiation, including *Ncoa2*, *Thy1*, *Hmgcr*, and *Il1r1*[26–29]. We then performed KEGG pathway analysis and found a clear Th17 cell differentiation signature in colonic OT-II cells vs colonic non-OT-II cells (Supplementary Data 5). In summary, we found that transferred OT-II cells adopt a Treg cell/Th2 signature in the duodenum, jejunum, and colon, whereas a Th17 signature is increased in the colon. These in vivo findings are consistent with our in vitro findings using DCs isolated from different regions of the gut to induce T-cell differentiation (Fig. 3a, b) and also consistent with Esterhazy et al. showing that lymph nodes (LN) draining distal regions of the gut favors the differentiation of RORγT+ Treg cells[17].

To investigate the role of intestinal LP DCs in T-cell differentiation in vivo, we performed kinetic experiments in which sorted naive T cells from OT-II mice were adoptively transferred to recipient congenic CD45.1 mice. Recipient mice received an intrarectal injection of OVA 16 h after T-cell transfer and were continuously fed with OVA in the drinking water throughout the experiment. Mice were then killed 24 h (1 day), 84 h (3.5 days), and 180 h (7.5 days) after cell transfer, and CD45.1 cells were analyzed from spleen, PP, lower gut-draining lymph nodes (iliac+caudal+cecum) (ccLN), mesenteric lymph node (mLN), uSI, and colon. As shown in Supplementary Fig. 4a, the majority of transferred cells were found in the spleen after 24 h and no cells were observed in the colonic LP. Moreover, we did not detect Foxp3 or RORγt staining at this stage in any transferred cells at any investigated sites. Conversely, transferred cells were found in all investigated sites at day 3.5 with no difference in frequencies among them, and they acquired an activated phenotype as shown by increased expression of CD44

(Supplementary Fig. 4b). In this case, Foxp3 and RORγt staining were only slightly detected in the spleen, mLN, ccLN, and PP, but were increased in cells sorted from the small and large intestine (Supplementary Fig. 4c, d). Accordingly, transferred T cells expressing Foxp3 at day 7.5 were also found preferentially in both small and large intestines as compared with other lymphoid organs. (Supplementary Fig. 4d). Interestingly, we found a trend toward an increased percentage of transferred T cells in the PP at day 7.5 as compared with the uSI, but these cells expressed lower Foxp3 than cells from the uSI. Taken together, these findings suggest that the gut is the preferential site for antigen-dependent T-cell differentiation and that intestinal DCs orchestrate T-cell differentiation.

**PD-L1+ and XCR1+ DCs function to maintain intestinal homeostasis.** RNA-seq and flow analysis showed that PD-L1 was highly expressed on LP DCs in the duodenum, whereas XCR1 was highly expressed on colonic DCs (Fig. 4a–c). PD-L1 is a ligand for the PD-1 that regulates cell activation and tolerance[30]. To investigate PD-L1+ DCs in the upper gut, we sorted duodenal and jejunal DCs from $CD274^{-/-}$ ($Pd\text{-}l1^{-/-}$) mice and performed RNA-seq. As shown in Fig. 4d, e, we found upregulation of *C3*, *C1qc,* and *C1qa* (genes related to complement), *Ccl24* and *Cxcl1* (chemokine related genes), and *Ecm1*, a gene encoding extracellular matrix protein 1 that is involved in inflammatory bowel disease[31,32].

XCR1+ DCs contribute to oral tolerance[9,33] and cancer immunity[34,35]. Little is known about the function of XCR1+ DCs in the intestinal LP. Because XCR1−/− mice are not commercially available, we investigated the role of XCR1-expressing DCs in the colonic LP by sorting XCR1− DCs from XCR1^DTA mice, in which diphtheria toxin A is specifically expressed in XCR1+ cells and allows depletion of XCR1+ cells[33]. We found that XCR1− DCs had higher expression of *Irf4*, which is associated with Sirpα+ cDC2s and decreased levels of *Irf8* (Fig. 4d, e), which is associated with XCR1+ cDC1s[36]. Furthermore, XCR1− DCs had increased expression of *Il1b*, *Cd80*, *Cd101,* and the C-type lectin receptors *Cd209a-d*, and decreased expression of *Itgae* (CD103), *Tlr3*, and *Clec9a* (Fig. 4d, e). Gene Set Enrichment Analysis (GSEA) showed that genes altered in both PD-L1− and XCR1− DCs were associated with inflammatory response pathways (Fig. 4f). We also found that supernatants from $Pd\text{-}l1^{-/-}$ DCs cultured with naive OT-II CD4+ T cells increased IFN-γ and TNF levels whereas supernatants from colonic XCR1− DCs increased Th17-related cytokines (IL-17A, IL-17F, Il-21, IL-22, and IL-6) (Supplementary Fig. 5a, b).

We then investigated the role of PD-L1+ and XCR1+ DCs in vivo. To specifically delete PD-L1 from DCs, we used $CD11c^{Cre}xPd\text{-}l1^{flox/flox}$ mice (Supplementary Fig. 6a). Of note, we did not observe altered Treg or Th17-cell distribution in the uSI or upper gut-draining lymph nodes from these mice and found no sign of inflammation in $CD11c^{Cre}xP\text{-}l1^{flox/flox}$ mice as compared with littermate controls under steady-state conditions (Supplementary Fig. 6b).

We then employed two well-known models of intestinal inflammation: 5-fluorouracil (5-FU)-induced mucositis, which primarily affects the small intestine[37], and DSS-induced colitis that affects the large intestine[38]. We found that $CD11c^{Cre}xPd\text{-}l1^{flox/flox}$ mice had increased inflammation induced by 5-FU, which was associated with a loss of mucosal architecture and significant crypt shortening (Fig. 5a). Consistent with this, we found increased neutrophil (Ly-6G high) infiltration into the small intestine LP of $CD11c^{Cre}xPd\text{-}l1^{flox/flox}$ mice after 5-FU treatment. (Fig. 5b; Supplementary Fig. 6c). Increased RORγt

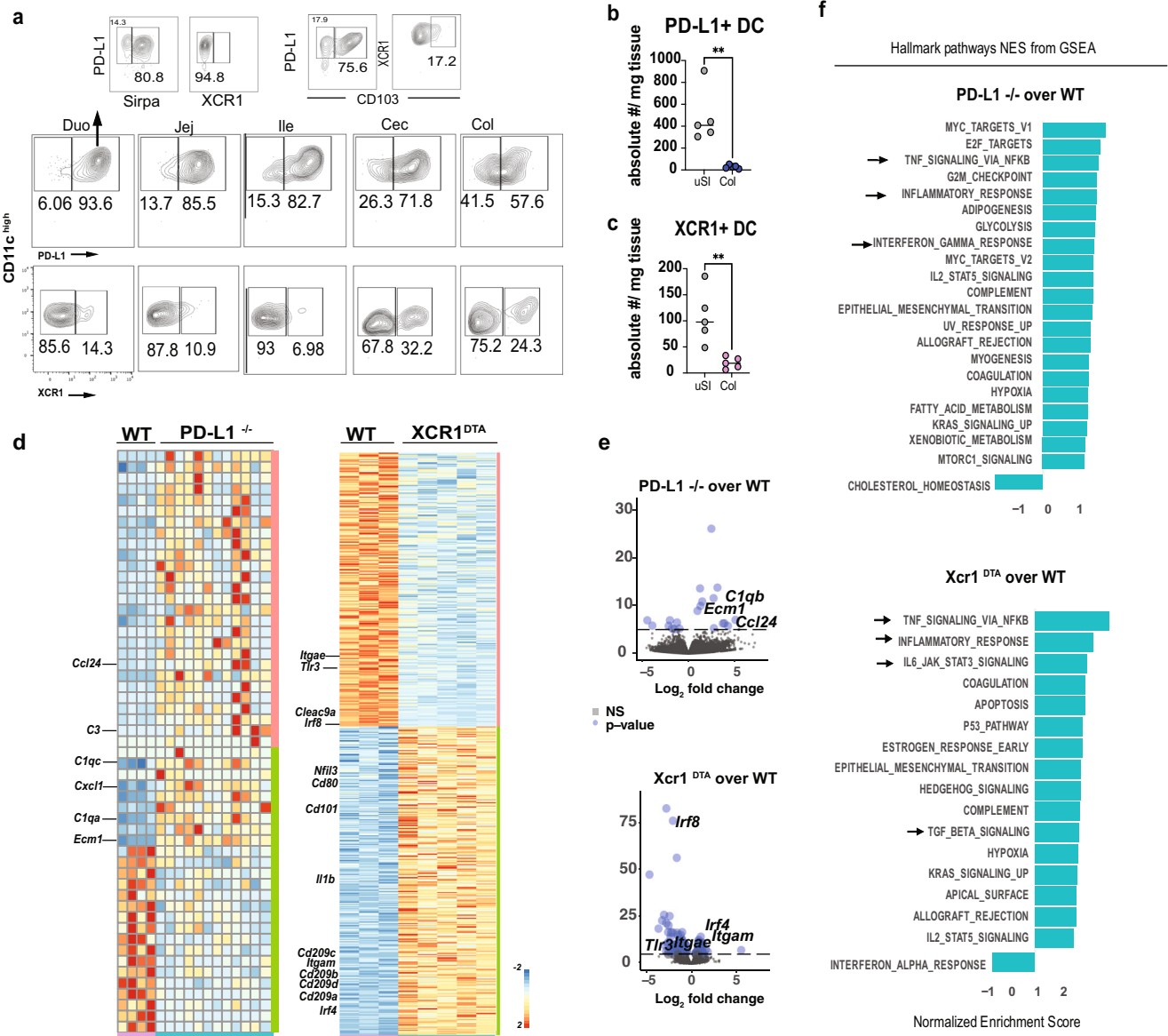

**Fig. 4 Pro-inflammatory signature of intestinal DC lacking PD-L1 or XCR1. a** Flow cytometry analysis of PD-L1+, XCR1+, Sirpα+, and CD103+ in LP DC from XCR1venus mice in five segments of the gut. **b** Absolute number of PD-L1+ and **c** XCR1+ DCs from C57BL/6 mice gated on singlets, live, CD45+, dump− (CD19, B220, CD90.2, F4/80, CD64) CD11c+ per gram of tissue upper Small Intestine (uSI) (duodenum and proximal jejunum) and colon (Col). $N = 5$ mice/group. Graphs represent mean ± SEM. Unpaired two-tailed Student's $t$ test. **$p = 0.003$ (PD-L1), **$p = 0.005$ (XCR1). **d–f** Heatmap (**d**), volcano plot (**e**), and Gene Set Enrichment Analysis (GSEA) pathways (**f**) of modulated genes found in RNA-seq analysis from DCs sorted from uSI LP of $Pd$-$l1^{−/−}$ and littermate controls, and colonic LP of XCR1DTA and littermate controls. $N = 3$ of a pool of $N = $ at least three mice over at least three independent experiments. Source data are provided as a Source Data file. Wild-type (WT); gene set enrichment analysis (GSEA).

expression as well as IL-17 and IFN-γ production, but not T-bet expression, was found in T cells from LP of $CD11c^{Cre}xPd$-$l1^{flox/flox}$ mice compared with controls (Fig. 5b; Supplementary Fig. 6c). Moreover, CD86 was upregulated in the uSI DCs of $CD11c^{Cre}xPd$-$l1^{flox/flox}$ mice after 5-FU treatment (Supplementary Fig. 6c).

DSS-induced colitis was exacerbated in XCR1DTA mice with increased inflammatory cell infiltrate, weight loss, and a worse DSS-colitis macroscopic score (Fig. 5c). Increased monocyte and neutrophil infiltration were found in the colonic LP of XCR1DTA mice as compared to littermate controls (Fig. 5d). We also found that IL-17, but not IFN-γ, production by LP T cells was increased in XCR1DTA mice. In this case, T-bet was decreased in colonic LP T cells (Fig. 5d; Supplementary Fig. 6c). Furthermore, we found

decreased LP Treg cells and increased RORγt+ T cells in XCR1DTA mice as compared with littermate controls in the DSS-induced colitis model. Thus, PD-L1+ and XCR1+ DCs play an important role in the small and large intestines, respectively, under inflammatory conditions. Of note, because PD-L1+ and XCR1+ DC are found in all intestinal compartments, we cannot rule out they limit inflammation beyond their site of enrichment.

We investigated duodenal and colonic biopsies from donors without inflammatory disease of the intestine (Supplementary Data 6). Human CD141+ (BDCA-3) and CD1c+ (BDCA-1) cDCs are analogous to mouse cDC1s and cDC2s[39–41]. Thus, we used these markers to identify DCs in human intestinal biopsies. Using flow cytometry, we found that CD1c+ and CD141+ cells were equally distributed in the duodenum and colon (Fig. 6a). Of

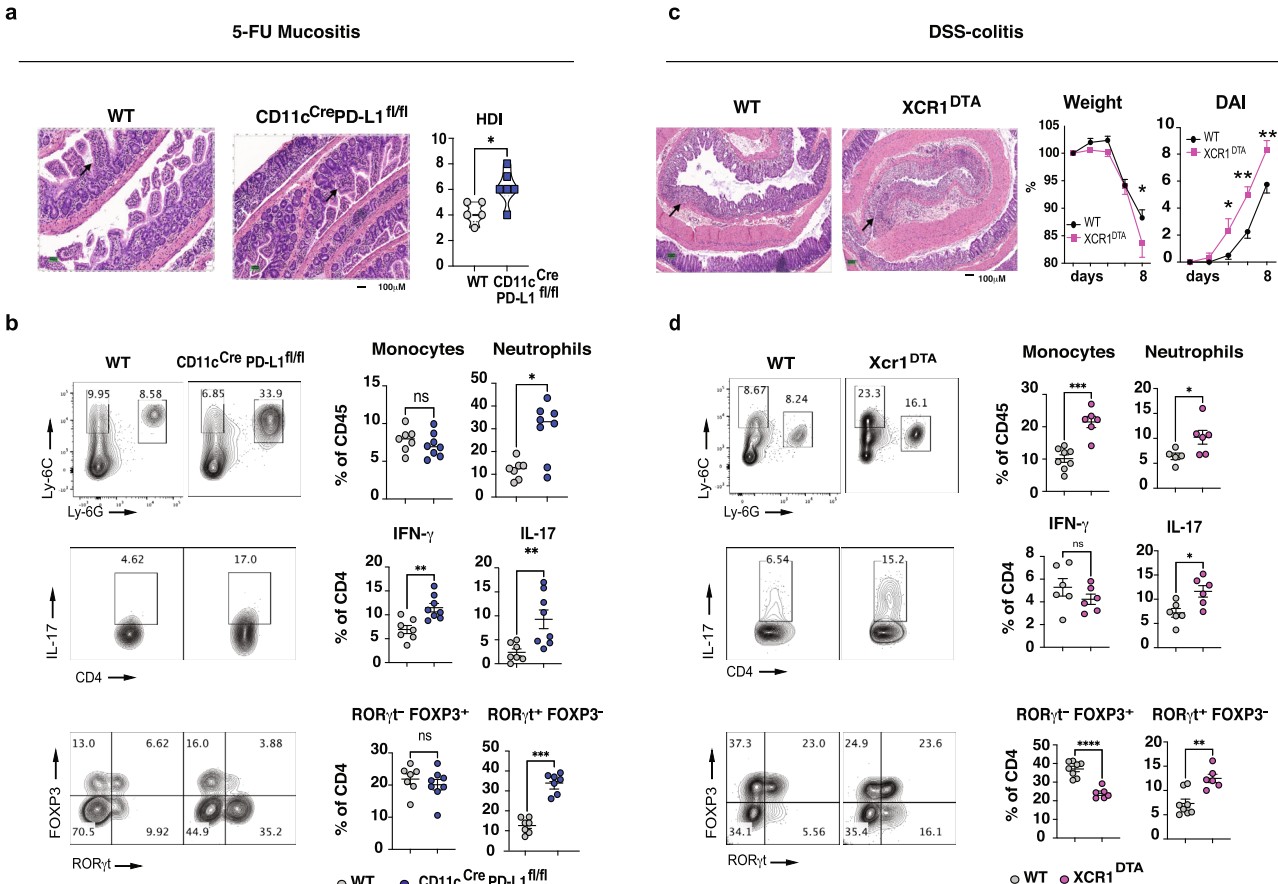

**Fig. 5 PD-L1⁺ and XCR1⁺ DCs are region-specific regulators of gut homeostasis. a** Histological damage index (HDI) and Hematoxylin and Eosin (H&E) staining from distal jejunum from littermate controls and *CD11c^Cre^xPd-I1^flox/flox^* mice after 72 h of 5-FU administration (10X); N = at least five *p = 0.032. **b** Flow cytometry representative dot plots and quantification of LP cells (monocytes, neutrophils, IL-17-producing CD4 T cells, Treg cells, and RORγt expressing CD4 T cells) isolated from upper small intestine (duodenum+jejunum) of *CD11c^Cre^xPd-I1^flox/flox^* mice and littermate controls. Unpaired two-tailed Student's t test *p = 0.01, **p = 0.002, **p = 0.007 (IL-17), ***p = 0.0006. **c** Weight loss *p = 0.041, Disease Activity Index (DAI) *p = 0.048, **p = 0.0015, **p = 0.0029 (day8) and H&E from the colon in DSS-treated littermate controls and XCR1^DTA^ mice (×10), N = 5. **d** Flow cytometry (FC) representative dot plots and quantification of LP cells as shown in **b** isolated from colonic LP cells from XCR1^DTA^ mice and littermate controls. Boxes and dots represent mean ± SEM. *p = 0.044, *p = 0.014 (IL-17), **p = 0.001, ***p = 0.0001, ****p < 0.0001. Non-parametric Student's t test with unpaired two-tailed Mann–Whitney post test was used for histological analysis, unpaired two-tailed Student's t test was used for FC and two-way-ANOVA followed by Sidak post hoc test for multiple comparisons was used for DAI and weight loss analysis. Experiments were performed independently in duplicate. Source data are provided as a Source Data file. Wild-type (WT).

note, we identify concomitant expression of CD1c and CD141 by immunofluorescence and flow cytometry staining indicating that these markers do not distinguish intestinal cDC1s and cDC2s in humans (Fig. 6a).

We then analyzed PD-L1 and XCR1 in CD11c⁺ cells from duodenal and colonic biopsies from the same individual by flow cytometry.

As we observed in mice (Fig. 4a, b), the frequency of PD-L1⁺ DCs was higher in duodenal DCs (Fig. 6b). However, we did not found differences in the frequency of XCR1⁺ DCs in the duodenum vs the colon (Fig. 6b) in human paired-up biopsies. The great variance of XCR1⁺ DC frequencies among subjects suggests a strong influence of microbiome in XCR1⁺ DC distribution in the gut.

## Discussion

It has been suggested that DCs are phenotypically distinct based on their anatomical location in the gut, and antigenic diversity throughout the intestine has been proposed as being responsible for this gut region DC specificity[42]. Esterhazy et al.[17] have shown that gut-draining lymph nodes are specific to the functional gut

segment to which they are associated and that the DC signature and polarization of T cells differ between segment-specific draining lymph nodes. After migration to the gut, DCs are subject to cues that can promote plasticity and functional specialization[43], and thus LP and lymph node DCs should be carefully distinguished.

Here, by means of phenotypical analysis, genetic manipulation, and functional characterization, we investigated DCs from the different intestinal LP compartments that are associated with tolerance to dietary antigens and to the microbiota (Supplementary Fig. 7a). We found that the LP DC number and frequency were significantly affected by the intestinal compartment. Classical cDC1s (CD103⁺CD11b⁻XCR1⁺) and cDC2s (CD103⁺ CD11b⁺Sirpα⁺) were differentially distributed throughout the gut and we found that DC subsets expressing PD-L1 and XCR1, were oppositely enriched according to the gut region. Consistent with this, when we reanalyzed Esterhazy et al.[17] RNA-seq data from gut region-specific draining lymph nodes (LN) we found that *Cd274* (PD-L1) was increased on DCs from duodenal LN whereas *Xcr1* expression was increased on DCs from cecal-colonic LN (Supplementary Fig. 7b, c).

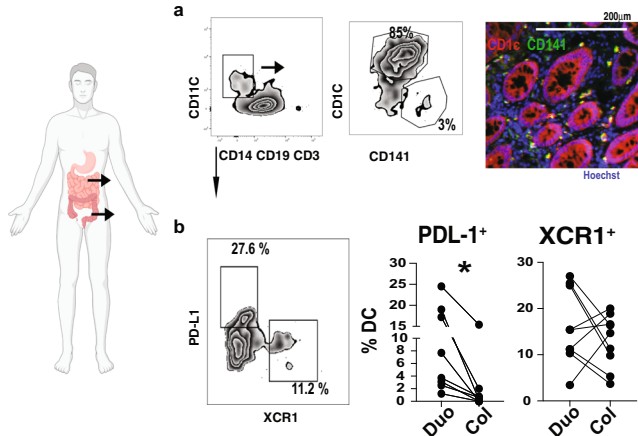

**Fig. 6 PD-L1+ and XCR1+ DCs isolated from duodenum and colonic human biopsies. a** Flow cytometry dot plots from duodenal and colonic human biopsies showing DC phenotype, and immunofluorescence showing CD1c and CD141 colocalization in colonic biopsies (20X). **b** Graphs represent the percentage of PD-L1+ and XCR1+ DCs gated on singlets, live, HLA-DR$^{High}$, dump$^-$ (CD19, CD3, CD14) and CD11c$^+$ cells; $N = 8$, paired two-tailed Student's $t$ test, *$p = 0.031$. Source data are provided as a Source Data file. Biorender was used for illustration.

The role of PD-L1+ DCs has been investigated in cancer because of their ability to suppress T-cell function[44] and it has been shown that PD-L1+ DCs acquire a regulatory profile (mregDCs) that impairs antitumor immunity[45]. The mregDC program is activated by canonical cDCs following uptake of tumor antigens, suggesting that changes within the gut environment may modulate PD-L1 expression by DCs[46]. We found that PD-L1 expression in duodenal cDC2s was higher than in colonic cDC2s showing regional distribution of PD-L1.

*Pd-l1*$^{-/-}$ DCs isolated from the upper gut upregulated the complement genes and imprinted a Th1-like phenotype to T cells in vitro. Accordingly, in the presence of C1q, DCs secrete more IL-12p70 and induce the production of IFN-γ by cultured T cells[47]. Furthermore, in the 5-FU-induced mucositis model, deletion of PD-L1 in DCs worsened inflammation in the small intestine, supporting a regulatory/homeostatic role of these cells in the gut.

Recently, Brown et al.[48] and colleagues have identified a novel cDC2 lineage defined by T-bet and RORγt expression in mice and humans. By reanalyzing their RNA-seq data we found that cDC2A and cDC2B in mLN also highly expressed PD-L1 (Supplementary Fig. 7d). Importantly, comparison between our RNA-seq and Esterhazy et al.[17] and Brown et al.[48] reanalyzed RNA-seq data demonstrated that DCs from intestinal LP were distinct from DCs found in draining LN including the correspondent gut region-specific LN (Supplementary Fig. 7e–g).

As opposed to the duodenal LP that had a higher frequency and number of PD-L1+ DCs as compared with colonic DCs, the colonic LP was enriched in XCR1+ DCs. It is possible that microbiota perturbation and dietary intervention lead to an alteration of XCR1+ and PD-L1+ DCs, consistent with the concept that both PD-L1 and XCR1 are important regulatory molecules that are modulated by the gut milieu. In humans, PD-L1 was significantly increased in duodenal vs colonic paired-up biopsies but we did not find differences in XCR1 expression. The discrepancy between humans and mice in terms of XCR1 could be related to differences in the microbiome.

In the absence of XCR1+ DCs, mice had increased colonic inflammation in the DSS-colitis model. Ohta et al.[33] have shown that mice lacking XCR1+ DCs have reduced intraepithelial and

LP T-cell populations, with the remaining T cells exhibiting an abnormal phenotype including increased IL-17a transcripts. Accordingly, we found increased Th17-related cytokine production by T cells cultured with colonic DCs from XCR1$^{DTA}$ mice. Of note, Th17 cell dysregulation can increase the risk of severe colitis[49] and RORγt inhibitors reduce gut inflammation in murine models of IBD[50]. Thus, the increased IL-17 production in the absence of XCR1+ DCs could explain in part the increased colonic inflammation observed in DSS-treated XCR1$^{DTA}$ mice.

In summary, our study identified LP DC subsets in distinct regions of the upper and lower intestine based on PD-L1 and XCR1 expression and showed that they play a critical role in gut homeostasis. Furthermore, our findings indicate that LN DCs cannot be considered as unique representatives of intestinal DC and thus, tissue specificity should be carefully considered in intestinal DC biology.

## Methods

**Mice.** Male and female mice were used. XCR1$^{DTA}$ mice were generated by cross-breeding B6.Cg-Xcr1<tm4(cre)Ksho> (RBRC09929) generously donated by Dr. Tsuneyasu Kaisho and B6.129P2-Gt(ROSA)26Sortm1(DTA)Lky/J (Jax 009669). XCR1$^{Venus}$ mice were generated by breeding B6.Cg-Xcr1<tm1Ksho> (RBRC09486) also donated by Dr. Tsuneyasu Kaisho (RIKEN, Japan). *PD-L1*$^{fl/fl}$ *CD11c*$^{Cre}$ mice were provided by Dr. Arlene Sharpe and *Pd-l1*$^{-/-}$ mice from Dr. Murugaiyan Gopal from BWH/HMS. Itgax reporter mice were generated by breeding Itgax-Cre mice and mT/mG mice (B6.129(Cg)-Gt(ROSA)26Sor$^{tm4(ACTB-tdTomato,-EGFP)Luo}$/J (Jax# 007676). S100a4 reporter mice were generated by breading B6.C-Tg(S100a4-cre)1Egn/JhrsJ (Jax 030644) and mT/mG mice. C57BL/6J mice (000664), CD45.1B6.SJLPtprca Pepcb/BoyJ (002014) and donor mice B6.Cg Tg(TcraTcrb) 425Cbn/J (004194) were purchased from the Jackson Laboratory. All mice used were C57BL/6 background. Mice were housed under specific pathogen-free conditions at Hale Building for Transformative Medicine at Brigham and Women's Hospital following ethical regulations for animal testing and research according to the animal protocol guidelines of Brigham and Women's Hospital Institutional Animal Care and Use Committee (IACUC). Light cycle from 7 am–lights on, 7 pm–lights off, the temperature of 68–75 F, the humidity of 35–65%. Mice purchased from the Jackson laboratory were acclimated in the local animal facility for at least 1 week prior to study initiation and genders were matched for each experiment. Otherwise specified, all mice used were 8–10 weeks old at the initiation of the study.

**Acquisition of human biopsies, storage, and LP isolation.** Human duodenal and colonic biopsies from the same patient were obtained at Biogastro in Belo Horizonte, Brazil through approved research (CAAE No. 35312820.7.0000.5149, COEP, UFMG) following good clinical practice guidelines and declaration of Helsinki. Consent forms were signed voluntarily by each patient prior to sample collection. All donors were free of chronic intestinal disease (Supplementary Data 6). Two to four 2 mm × 2 mm biopsies per area were collected using biopsy forceps from the duodenum and proximal colon. Duodenal and proximal colon biopsies were collected in T-cell media (RPMI 1640 medium with Glutamin without NaHCO$_3$ (Gibco), 10% fetal bovine serum (FBS; Cultilab) gentamicin (50 µg/ml) (Gibco), non-essential amino acids (NEAA; 0.1 mM) (Sigma-Aldrich), 1 mM sodium pyruvate (Gibco)) at final concentration and were stored on ice until cryopreservation. Fresh biopsies were then slow-frozen in 1 mL of freeze medium (10% dimethyl sulfoxide) (Sigma-Aldrich) and 90% FBS and promptly placed in a Nalgene Mr. Frosty freezing container (Sigma-Aldrich) to freeze at a 1 °C per minute cooling rate in a −80 °C freezer according to the protocol[51]. For LP cell isolation, cryovials containing frozen biopsies or surgical tissue were quickly thawed at 37 °C, and biopsies were transferred from T-cell media to media for tissue digestion as described in the LP cell isolation section.

**16S rDNA sequencing and analysis of gut microbiome.** Fecal microbiota samples were collected from duodenum, jejunum, ileum, cecum, and colon of 8-week old naive C56Bl/6 J mice after at least 1 week of acclimatization post mouse delivery. QIAamp Fast DNA Stool Mini Kit (Qiagen, cat#12855) was used DNA extracting. Amplicons spanning variable region 4 (V4) gene were generated with primers and barcodes (515 F, 806 R)[52] using HotMaster Taq and HotMaster Mix (QuantaBio) and paired-end sequenced using Illumina MiSeq platform. The sequence was performed at the Harvard Medical School Biopolymer Facility. Established protocol for QIIME 2 software[52] and LEfSe (Linear discriminant analysis Effect Size) version 1[53] was used for data analysis. Sequences were quality filtered in which reads were truncated if two consecutive bases fall below a quality score of Q20 (1% error), and reads that were <75% of full length were discarded[54].

**LP cell isolation**. LP extraction was performed according to Moreira's protocol[38] with few modifications. In brief, feces and mucus were removed from the collected intestines. Tissue was flipped (inside out) and washed several times using Hank's Balanced salt solution (HBSS) (Gibco) (without MgCl₂, CaCl₂, and MgSO₄) solution supplemented with 5% FBS (Gibco) and 25 mM 4-(2-hydroxyethyl)-1-piperazineethanesulfonic acid (HEPES; Lonza). After cleaning, the entire gut segment was incubated for 20 min at 37 °C in an orbital shaker with HBSS supplemented with 15 mM HEPES (Sigma) + 5 mM ethylenediaminetetraacetic acid (EDTA) (Invitrogen) 10% FBS + 0.015% or 1 mM 1.4-dithiothreitol (DTT; Roche). After incubation, tissues were washed two times with HBSS-EDTA solution supplemented with 25 mM EDTA. Tissues were chopped into smaller pieces and incubated with exVIVO (Lonza) media containing Liberase TL (Roche) 5 mg/80 ml + Dnase 30 mg/ml, grade 1 (Sigma-Aldrich) for 45 min in a 37 °C shaker. The supernatant containing LP cells was meshed using a 100 μm and 70 μm cell strainer and washed with HBSS solution supplemented with 2 mM EDTA 5 mM HEPES and 5% FCS. The cell suspension was then purified using CD45 microbeads according to the manufacturer's protocol (Miltenyi Biotec). The sorted cell suspension was resuspended in FACS buffer and stained for flow cytometry.

**Mass cytometry**. Procedures were performed according to David's protocol[55] with few modifications. In brief, cell suspensions were washed in cytokine stabilization (CSB) buffer once (low-barium PBS without MgCl₂, CaCl₂ (Gibco), bovine serum albumin protease-free (Sigma-Aldrich), and sodium azide (Sigma-Aldrich) and plated in 96-well polypropylene plates. Cells were then incubated for 5 min in RT with a cisplatin solution (1:1000) (Cell-ID Cisplatin; Fluidigm) for cell viability staining. Cells were washed and incubated with 20 ml of Fc-Block antibodies (rat-anti mouse CD16/32; clone 2.4G2; 1:100; BD Biosciences) for 10 min at RT. Later, 20 μl of metal-coupled surface antibody cocktail was added to each well (see Data 3 and 4). After 30 min of incubation, cells were washed and resuspended in 1.6% PFA solution (Pierce 16% formaldehyde (w/v), methanol-free) (Thermo Fisher) for 10 min. Cells were then washed in CSB buffer and resuspended in intercalator solution (Cell-ID Intercalator-Ir500 μM in Maxpar Fix and Perm Buffer; Fluidigm) for overnight incubation 4 °C. On the next day, cells were washed in either MilliQ water (low barium) or CAS water (Fluidigm) once and then eluted in 200 μl MilliQ water containing 1:10 dilution of EQ beads (EQ Four Element Calibration Bead; Fluidigm) for normalization. Cells were filtered and acquired in CyTOF on Helios platform. Analysis of CyTOF data was performed using either Cytobank (version 6.2)[56] or Flowjo (Tree Star Inc) versions 10.5.3 and 10.6.2.

**Smart-Seq2 RNA-seq and data analysis**. Sorted cells were collected in RNA-free Eppendorf tube containing 5 ml of TCL buffer (Qiagen) and 1% 2-mercaptoethanol (Gibco). After sorting, cell lysates were transferred to Eppendorf twin.tec PCR 96-well plate (Eppendorf) and sealed with Microseal 'F' Foil (Bio-Rad Laboratories, Inc.). The plate was stored at −80 °C until transportation to Broad Genomics Platform. Lysate cleanup and reverse transcription of mRNA, transcriptome amplification and PCR cleanup, Nextera XT sequencing-library construction, DNA SPRI bead cleanup, and 2 × 38 bp paired sequencing were performed according to Trombetta protocol[57]. Sequencing was run on al Illumina NextSeq500 at the Broad Genomics platform, Broad Institute of MIT and Harvard. Sequencing data were demultiplexed and provided by the Broad Institute in FASTQ format. Reads were quantified at the transcript level using Salmon[58,59] against an Ensembl catalog, and aggregated to the gene-level using tximport[59]. Differential analysis was performed using DESeq2[60] with a false discovery rate cutoff of 5%. Gene groups for GSEA were selected from MsigDB (https://www.gsea-msigdb.org/gsea/msigdb). For comparison against the[17] (GSE121811) and[48] (GSE130201) gene-level data, raw gene counts resulting from each experiment were compiled, normalized using the median of ratios method via DESeq2, and filtered for low abundance (mean normalized count cutoff of 5).

**Immunofluorescence and immunohistochemistry**. Duodenal and colonic samples were fixed in 4% paraformaldehyde (PFA) for 24 h and transferred to 70% ethanol until staining. Immunofluorescence from mouse tissue was performed by Specialized Histopathology Core at the Brigham and Women's Hospital/Dana-Farber. Multiplex immunofluorescent staining was performed on the Leica Bond RX automated staining platform using the Leica Biosystems Refine Detection kit (Leica, cat#DS9800). Anti-E-cadherin antibody (4A2; 1:100; Cell Signaling Technology) was labeled with Alexa Fluor 488 (Thermo Fisher) with citrate antigen retrieval. Anti-CD11c (D1V9Y; 1:350; Cell Signaling Technology) was labeled with Alexa Fluor 647 (Thermo Fisher) with citrate antigen retrieval. Counterstain was performed with Nucblue Fixed Cell Stain Readyprobes (Hoechst dye; Thermo Fisher). A custom ImageJ macro code was used to batch process images. In brief, the code applies a "rolling ball" background subtraction (with radius 30 pixels) and gaussian blur filter (radius 2 pixels), then uses threshold algorithms to generate mask image of channel CD11c. ImageJ "Analyze Particles" tool was used to analyze puncta statistics including size and number.

Immunofluorescence staining for human specimens was performed by Applied Pathology Systems that also provided the duodenum and colon samples for the study. Paraffin sections were dewaxed, rehydrated, and subjected to antigen retrieval in the microwave with AR9 Buffer (AR900, Akoya). Slides were blocked

with Blox All blocking buffer and 2.5% horse serum respectively (both from Vector Laboratories) prior to the incubation of primary antibodies. For CD141 and CD1c co-staining, anti-CD141 (E7Y9P; 1:1000; Cell Signaling Technology) and anti-CD1c (EPR23189-196; 1:250; Abcam) were used as primary antibodies, and AF594 (ab150080) and AF647 (ab150075) (both from Abcam) were used as secondary antibodies.

Immunohistochemistry staining was performed by ServiceBio/iHisto. In brief, slides were embedded in EDTA solution that was heated and reheated multiple times in a microwave. Slides were then incubated with 10% H₂O₂ at RT for 15 min. Later, slides were incubated with BSA for 20 min (RT). Primary antibody CD11c (D1V9Y; 1:100; Cell Signaling Technology) was applied overnight at 4 °C and secondary antibody IgG (H+L) Cross-Adsorbed Goat anti-Rabbit, horseradish peroxidase (1:2000; Thermo Fisher) was applied for 50 min (RT). Slides were then incubated with DAB buffer (Thermo Fisher) at 1:10 dilution. The color was differentiated in 1% HCl alcohol solution for 7–10 s and 0.6% ammonium solution was applied to blue the nuclei. Slides were dehydrated and mounted. A custom ImageJ macro code was used to batch process images. In brief, the code uses "Color Deconvolution" plugin (Gabriel Landini https://imagej.net/Colour_Deconvolution) option "H DAB" to generate "Color_2" channel as "Brown" region. The channel is filtered with "Gaussian" filter (radius 1) and segmented with "Triangle" thresholding methods to achieve mask of "brown" region.

**Flow cytometry**. Unless otherwise specified, antibodies were purchased from Biolegend. Fluorescein isothiocyanate-conjugated (FITC) or Alexa Fluor 488 monoclonal antibodies (mAbs) to Foxp3 (MF-14; 1:100), CD25 (3C7; 1:100), CD209b (LWC06; 1:150), Ly-6C (Hk1.4; 1:100) phycoerythrin (PE)-conjugated mAbs to CD11c (N418; 1:250), CX3CR1 (SA011F11; 1:250); PerCP-Cy5.5-conjugated mAbs to Nk1.1 (PK136), B220 (RA3-6B2), Ly-6G (1A8), CD3e (145-2C11), CD19 (6D5) (all at 1:300, BD Biosciences); Allophycocyanin (APC)-conjugated mAbs to CD45 (30-F11; 1:400), TLR3 (1F8; 1:250), IFN-γ (XMG1.2; 1:200; Thermo Fisher), RORγt (B2D; 1:200; Thermo Fischer), CD86 (GL-1; 1:400), CD103 (2E7; 1:400; Thermo Fischer); IFN-γ mAbs conjugated to BUV395 (TC11-18H10; 1:250; BD Bioscience); RORγt mAbs conjugated to Brillant Violet (BV) 421 (Q31-378; 1:400; BD Bioscience); CD45 mAbs conjugated to Alexa Fluor (AF) 700 (30-F11; 1:350; Thermo Fisher); APC-Cy7-conjugated mAbs to CD19 (6D5; 1:300), B220 (RA3-6B2; 1:300), CD90.1 (53-2.1; 1:300), CD62L (MEL-14; 1:300); PE-Cy7-conjugated mAbs to CD44 (IM7; 1:300), CD101 (Moushi101; 1:150; Thermo Fisher), CD172/Sirpa (P84; 1:200), IL-17A (eBio17B7; 1:200); BV421-conjugated mAbs to Siglec-f (E50-2440; 1:400; BD Bioscience), CD274/PD-L1 (10 F.9G2, 1:200); Ly-6G (1A8; 1:350, BD Bioscience) Il-17A (TC11-18H10; 1:200, BD Bioscience), BV711-conjugated mAbs to CD8 (53-6.7; 1:300; BD Bioscience) and CD103 (M290; 1:300; BD Bioscience); BV786-conjugated mAb to CD103 (M290; 1:300; BD Bioscience); CD4 mAbs in AF 700 (GK1.5) and BV605 and BV786 (RM4-5; 1:400; BD Bioscience). Pe-dazzle mAbs to Ly-6G (1A8; 1:500) T-bet (4B10; 1:200)—Alexa Fluor 488 anti-GFP mAb was used to amplify signal when reporter mice was used (FM264G; 1:100). Surface staining was performed according to standard procedures at a density of 0.3–1 × 10⁶ cells per 50 μl, and volumes were scaled up accordingly. Fc block (rat-anti mouse CD16/32; clone 2.4G2; BD Biosciences) was used at 1:100 dilution. CountBright Absolute Counting Beads (Thermo Fisher) was used according to manufacturer's protocol to obtain absolute cell number from gut compartments. Flow cytometry sample acquisition was performed in BD Fortessa, BD Symphony and FACScanto (BD Biosciences). FlowJo software (Tree Star Inc) was used for analysis. For Foxp3 intracellular staining, Foxp3 staining kit (Thermo Fisher, cat# 00-5523-00) was used as per manufacturer's instructions. For cytokine staining, mouse intracellular cytokine staining kit (BD Biosciences, cat#554714) was used. Fluorescence minus one was used as a parameter to identify gating strategy. Cells isolated from human biopsies were incubated with Human Fc Block (1:100; BD Bioscience) and stained with anti-human HLA-DR APC-Cy7-conjugated mAb (L243; 1:300; Biolegend), PE-conjugated mAb to CD11c (B-ly6; 1:200; BD Bioscience), PerCP-C5.5-conjugated mAbs to CD3 (UCHT1), CD19 (HIB19) and CD14 (M5E2) (all at 1:300; BD Bioscience); PE-Cy7-conjugated mAb to CD1c (L161; 1:300; Biolegend), FITC-conjugated mAb to CD274/PD-L1 (MIH2; 1:100; Biolegend), APC-conjugated mAb to CD141 (1A4; 1:300; BD Bioscience), BV421-conjugated mAb to XCR1 (S15046E; 1:200; Biolegend).

**RT-qPCR**. DCs were presorted using CD11c beads and sorted using flow cytometry. DCs were determinate as Live, CD45⁺; T, B cells (CD90.2-, CD19-, B220-) and macrophage (F4/80-, CD64-) were dumped; CD11c high population was sorted. Flow cytometry cell recovery and purity were higher than 97%. DCs were sorted from different intestinal compartments directly onto lysis buffer and RNA was extracted using PicoPure RNA Isolation Kit (Applied Biosystems, cat# KIT0204) following the manufacturer's instructions. RNA expression was accessed by TaqMan Gene Expression Assay (FAM) (Thermo Fischer). *S100a4* (Mm00803372_g1), *Xcr1* (Mm00627650_m1), *Cd274* (Mm03048248_m1), *Cd209b* (Mm00499588_g1), *Actb* (Mm02619580_g1) were used. Cycles of housekeeping genes were satisfactory. Differentially expressed genes were identified by two-tailed unpaired Student's t test and statically relevant results consisted in p value < 0.05.

**Cell sorting and in vitro culture assays**. DCs from different intestinal LP regions were obtained by pooling gut segments from at least three to four mice/sample. LP DC cell suspensions were presorted using mouse CD11c MACS beads (Miltenyi Biotec) and then stained with viability dye (fixable viability dye eFluor 506; 1:1000; Thermo Fisher), APC-conjugated mAb to CD45 (30-F11; 1:400) BV605-conjugated mAb to F4/80 (BM8; 1:300) and CD64 (X54-5/7.1; 1:300), APC-Cy7-conjugated mAb to CD19 (6D5; 1:300), B220 (RA3-6B2; 1:300), CD90.1 (53-2.1; 1:300) and PE-conjugated mAb to CD11c (N418; 1:200) and sorted using FACS ARIA III (BD Biosciences) at the Flow Cytometry Core facility at the ARCND/BWH. For RNA-seq, sorted cells were collected in RNAse-free Eppendorf tube containing 5 μl of TCL buffer (Qiagen) and 1% 2-mercaptoethanol (Gibco). For RT-qPCR cells were sorted in lysis buffer provided in PicoPure RNA Isolation Kit (Applied Biosystems, cat# KIT0204). After collection cells were snap-freeze. For in vitro culture DCs were sorted into 96-well plates containing media Iscove's Modified Dulbecco's Medium + Glutamax-I + HEPES and NaHCO₃ (Gibco) supplemented with NEAA (MEM) (Gibco) 2-Mercaptoethanol (Gibco) and pen/strep. After sorting, cells were gently centrifuged, and media was changed.

T cells were presorted using mouse CD4 MACS beads (Miltenyi Biotec) and further stained for viability dye eFluor 506 (1:1000; Thermo Fisher), PerCP-Cy5.5-conjugated mAb to CD4 (RM4-5; 1:300) APC-Cy7-conjugated mAb to CD62L (MEL-14; 1:250), PE-Cy7-conjugated mAb to CD44 (IM; 1:250) and FITC-conjugated mAb to CD25 (3C7; 1:100). The purity of transgenic CD4+ T cells was verified by flow cytometry by staining an aliquot of cells with PE-Cy7-conjugated mAb to CD45.1 (A20; 1:300), APC-conjugated mAb to Vα2+ (B20.1; 1:400) PE-conjugated mAb to Vβ5+ (MR9-4; 1:300). Unless otherwise specified, all antibodies were purchased from Biolegend. T cells were sorted into Falcon tubes and then transferred to 96-well plates containing DCs. DCs and T cells were plated in the presence of OVA peptide₃₂₃₋₃₃₉ (InvivoGen). At least $1 \times 10^4$ DCs were plated in each well. Supernatants were collected for cytokine quantification by Legendplex (13-plex pro-inflammatory panel; Biolegend) at day 7 and cells were used for intracellular staining of Foxp3, ROR-γt, IFN-γ, IL-17A, and IL-10 (see Flow cytometry section).

**Adoptive T-cell transfer and intra-colonic OVA treatment**. Naive CD4 T cells from spleen and lymph nodes were isolated from 8-week-old OT-II CD45.1 mice and presorted using anti-CD4 MACS beads (Miltenyi Biotec). Naive cells (Live CD4⁺CD62L⁺CD44⁻CD25⁻) were then sorted using FACS ARIA III (BD Bioscience) and washed with PBS twice. The purity of transgenic CD4+ T cells was verified by flow cytometry (CD45.1+Vα2+Vβ5+). In all, $3 \times 10^6$ cells were transferred by tail vein injection. Mice had access to drinking water containing 8 g/L of OVA (Sigma-Aldrich) for 7 days. After 24 h of T-cell transfer, mice received intra-colonic injections of OVA (total of 2.4 mg; Inject Ovalbumin; Thermo Fischer) in three distinct colonic regions using a Karl Storz endoscopic system (see supplementary movie 1). For intra-colonic injections, mice were anesthetized using isoflurane inhalation and injections were carried out in the mucosal space using a custom injection needle (Hamilton, 33 gauge, small Hub RN NDL, 16 inches long, point 4, 45-degree bevel, part number 7803–05), a syringe (Hamilton Inc., part number 7656-01), a transfer needle (Hamilton Inc., part number 7770-02), and a colonoscopy with integrated working channel (Richard Wolf 1.9 mm/9.5 French pediatric urethroscope, part number 8626.431). After 7 days, mice were killed, and the upper gut (duodenum and proximal jejunum) and colon were harvested for T-cell recovery.

**DSS-induced colitis, 5-FU-induced mucositis, and histopathology**. Colonic inflammation in XCR1ᴰᵀᴬ and littermate controls was induced by the addition of 2.5% (w/v) DSS (40 kDa, MP Biomedicals) in the drinking water, for 7 days followed by 2 days of water offering, until euthanasia (day 9). The naive group received drinking water only. Mice had unlimited access to food during the experiment and were weighed daily for weight management assessment. DSS consumption was measured and changed every 2 days. For the mucositis model, CD11cᶜʳᵉxPd-l1ᶠˡ/ᶠˡ mice and littermate controls received an i.p. injection of 450 mg/kg solution of 5-FU (Sigma-Aldrich) in PBS. After 72 h, mice were euthanized, and the small intestine was collected. Samples from the colon and small intestine were fixed with a PBS solution containing 4% paraformaldehyde and then embedded in paraffin. In all, 5 μm of paraffin blocks were cut and stained with Hematoxylin and Eosin. For histological inflammation scoring, colon tissues were observed under a microscope, and sections were scored blindly by an expert pathologist at Harvard Medical School. DSS-histological scoring was based on a semi-quantitative scoring system considering inflammatory infiltrate and edema[38]. The disease activity index was calculated by the following formula: weight loss score + diarrhea score + rectal bleeding score. The mucosal damage caused by mucositis was assessed by six different parameters: reduction of the intestinal crypts and villi; disruptions and abscess formation in the crypts; thickening of the outer muscle layer; integrity of the epithelium and muscular layer; inflammatory cell infiltration; vacuolization and edema. Scores for each parameter ranging from 0 to 3 (0 = normal; 1 = mild; 2 = moderate; 3 = severe) were totaled to reach a histological damage index maximum of 18[61].

**Statistical analysis**. Results are shown as the mean values (±SEM) and considered statistically significant when comparisons between groups, using two-tailed Student's t test, one or two-analysis of variance (ANOVA) p values were <0.05. For histological scores, we used medians and non-parametric Student's t test with Mann–Whitney post test. Non-parametric ANOVA Kruskal–Wallis was used for microbiota studies and paired Student's t test was used for human data analysis. *p < 0.05 p < 0.05 **p < 0.01***p < 0.001 ****p < 0.0001. All experiments were performed at least two times to assure rigor and reproducibility.

**Reporting summary**. Further information on research design is available in the Nature Research Reporting Summary linked to this article.

## Data availability
Microbiome and RNA-seq data have been uploaded to the National Center for Biotechnology Information (NCBI) in the short read archive (SRA) under the BioProject primary accession code PRJNA733716. Batch analysis of intestinal DC using ImageJ code was deposit in Zenodo. DC database from (Esterhazy et al., 2019) (GSE121811)[17] and (Brown et al., 2019) (GSE130201)[48] were used to compare with LP RNA-seq data. All other relevant data supporting the key findings of this study are available within the article and its Supplementary Information files or from the corresponding author upon reasonable request. Source data are provided with this paper.

## Code availability
Batch analysis of intestinal DC using ImageJ code can be found at https://doi.org/10.5281/zenodo.4876494.

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

## Acknowledgements

Special thanks to Dr. Tsuneyasu Kaisho for the generous donation of B6.Cg-Xcr1<tm1Ksho> and B6.Cg-Xcr1<tm4(cre)Ksho> and RIKEN for mouse transferring support. We thank Dr. Arlene Sharpe for the generous donation of *Pd-l1*<sup>fl/fl</sup> CD11c<sup>Cre</sup> mice and Lucien Garo for *Pd-l1*<sup>−/−</sup> mouse breeding. We also thank Michael Blanchard from Neurobiology Imaging Facility at HMS for images assistance and Lai Ding from NeuroTechnology studio at BWH for tissue IF and IHC analysis support. Our thanks to Daniel Frederick from Fluidigm for technical support with CyTOF panel design and data analysis. We thank Rajesh Krishnan and Deneen Kozoriz from Flow Cytometry Core facility at ARCND-BWH as well as Eric Hass and Nicole Paul from Dana-Farber CyTOF core. For experimental support with human biopsies, we thank Pedro Henrique Prazeres from UFMG and Caroline Almeida de Lima from Clínica Biogastro, Brazil. We especially thank Weiner's Lab members: Shirong Liu, Abou El Hassan, Julia Rocha, Dvora Ghitza, and Stephen Rubino for their scientific support. We thank Gustavo Menezes and Maisa Antunes from UFMG, Brazil, for their assistance with immunofluorescence experiments. We also thank Daniel Mucida and Maria Cecilia Canesso from Rockefeller University for the scientific discussion and support for this work. T.G.M. was supported by Foundation of Neurologic Disease and is being currently supported by Susan Furbacher Conroy Fellow. D.M. is supported by a postdoc mobility fellowship from the Swiss National Science Foundation (P400PB_183910). M.A.O and A.M.C.F are supported by Conselho de Desenvolvimento Cientifico e Tecnologico (CNPq, Brazil). B.N.N. was supported by CAPES, Brazil. A.C.A. is a recipient of the Brigham and Women's Hospital President's Scholar Award.

## Author contributions

T.G.M. designed the project and experiments, carried out the experiments, and wrote the manuscript. D.M. performed intra-colonic and intravenous injections helped with Legendplex experiments, wrote the manuscript, and discuss experiment design. L.C. performed microbiome analysis and helped discuss experiments. J.L. performed RNA-seq analysis from Smart-2-seq experiments. E.L.C.L. helped maintain mouse breeding and carried out experiments. M.A.O. performed LP cell isolation from human biopsies. C.D.G. performed RNA-seq analysis and RNA-seq comparison–GEO Dataset, B.N.N. helped perform experiments. V.W. and A.S. helped in LP experiments, microbiome DNA isolation, and library preparation. L.G. helped in LP experiments and mouse breeding. D. C.A.L. collected intestinal human biopsies. M.G. provided *Pd-l1* full knockout mice and inputs for the manuscript. O.B., G.G., and A.C.A. provided input for experiments and helped correct the manuscript. R.M.R. assisted with the project and experiments and wrote the manuscript. A.M.C.F. designed the project and corrected the manuscript. H.L. W. supervised the project, designed experiments, and corrected the manuscript.

## Competing interests

The authors declare no competing interests.
