## [Peer Review File · Nature Communications]

REVIEWER COMMENTS

Reviewer #1 (Remarks to the Author):

General comments:

This is a tremendous amount of work, much of it of high quality, that digs into the very complex question of how DC are influenced by their anatomical location, and whether this in turn impacts the "flavour" of the T cell response. Assessing these DC in different challenge situations (5-FU, DSS) and environmental changes (Abx, AA diet) was then used to ascertain how these DC adapt to various perturbations.

Due to the broad scope of the study, the paper is quite dense and sometime hard to follow with some internal inconsistencies in the data and problematic choices of mouse models. It would be far better if the manuscript were reduced in scope and authors shored up their findings on the first 5 figures.

For example, if one were to simply focus on the anatomical differences in DC representation (an important finding in and of itself), there are some inconsistencies. Specifically, in Figure 4 authors state "These in vivo findings are consistent with our in vitro findings using DCs isolated from different regions of the gut to induced T cell differentiation". However, this is not born out in the data: T cells stimulated in vitro with colonic DCs had an elevated Foxp3/Il17 ratio (Fig. 4a-b). This is in opposition to the in vivo observation that OTII-specific T cells adopt a Th17 phenotype in the colon (Fig. 4c-e). In addition in Figure 5b-c the conclusion that there are greater numbers of XCR1+ DC in the colon per gram of tissue than PDL1+ DC does not compute with the representative flow cytometry plots displayed (eg, based on the frequencies of the representative plots in the colon, where PDL1+ DCs and XCR DCs make up 58% and 24% of CD11c-hi cells, respectively one would expect the opposite). Furthermore, the total number of DCs in the SI should be much larger than in the colon per g/tissue based on data in Fig. 1.

There could be many reasons for these inconsistencies. For example, while DC are used in culture, macrophages make up a significant portion of APCs in the gut. These are not accounted for in the in vivo experiments. Moreover, physiological differences in the DC-T cell ratio (lower in the colon compared to the SI) might account for in vitro vs in vivo discrepancies. Does titrating down the DC:T cell ratio in vitro skew the T cells towards being more Th17 rather than regulatory? In these cultures, how was the ratio of DCs and T cells rationalized for the small intestines and the large intestines? In the mouse colon, the actual T cell to APC ratio is far greater than the mouse small intestines.

In addition, where inflammatory insults are introduced (DSS, 5-FU) it is important for the authors to evaluate the immunosuppressive function of XCR1+ cDC1 and PDL1+ cDC2 across both the uSI and the LI. As it stands, these analyses are incomplete. In general, the description of these mouse lines is also lacking. For example, the use of the CD11c-Cre is problematic – this will also delete PDL1 on CD11c+ macrophages. Authors need to fully characterize the expression of PDL1 on various APC subsets (including macrophages) and/or use a different more cDC-specific Cre animal.

In summary, the authors are urged to reconsider how to best "package" this work, perhaps developing the AA diet (which is very interesting!) and Abx perturbations for another study and focusing on shoring up the aforementioned issues in Figures 1-5. I believe that when tidied up, this more limited data set will be an important resource for the community and lays the groundwork for a separate study on diet and microbiome perturbations for the future. Additional specific comments are below:

Other comments that need to be addressed:

Figure 2a: Interesting that Notch2 is upregulated in colonic DCs, as it's exclusively expressed by cDC2 (which are shown not to be the dominant DC population in this area). K. Murphy and colleagues show that these cDC2 are required for generating immune responses against C.

rodentium in the SI. Please comment on this.

Figure 2d: Why not also include levels of Sirpa (cDC2) in the RNA-seq data to assess changes across the different gut anatomical locations? This would provide a nice internal control validating the concept that cDC2 are more dominant in the SI vs the LI.

Figure 3a: Not intuitive how to interpret these FlowSOM plots. For example, CD209a (DC-SIGN) are previously shown to be down-regulated from the SI to the LI. But there are more red circles in the colon compared to the duodenum? Likewise, XCR1 in green is more visible in the duodenum than the colon. These data run counter to those in Figure 2D.

Figure 4a: It is appreciated that the cytokines measured here are often difficult to detect by Flow Cytometry, hence the use of ELISA. However, some of these cytokines are produced both by the T cells and the DCs themselves. I would recommend either showing the cytokine levels from DC co-cultured with CD4+ T cells that are not OVA-specific as a control or sorting the T cells post-culture and performing qPCR.

Figure 4b: Please provide dot plots for Foxp3 and IL17 for completeness.

Supplementary Fig. 3: Cytokines that were examined in both S3a and S3b should share the same Y axis for readers to make comparisons.

Figure 5J: These data are not convincing. Quantification (#DC per μm^2 or some other metric) is required.

Figure 6a-g: Were XCR1-expressing cDC1s in either the uSI or the LI perturbed by the AA-diet similar to PDL1-expressing DCs?

Figure 6h-n: Abx treatment definitely induces perturbations in DCs of the small intestine and should be similarly examined as the large intestine.

Figure 6 (general): RNA-seq analysis should be performed looking at Sirpa+ cDC2 clusters and XCR1+ cDC1 clusters individually, rather than all DCs together. Brown et al. had previously established that Abx treatment reduces the abundance of Tbet-expressing cDC2, which are functionally described to be immunoregulatory, in both the SI and the LI. Since the authors describe XCR1+ cDC1 to be the main immunosuppressive DC subset in the LI, the authors should distinguish whether the Abx treatment is impacting XCR1+ cDC1 OR cDC2 or both to possess a pro-inflammatory gene signature. The DC-T cell co-culture assays should also be performed with Abx-treated cDC1 OR Abx-treated cDC2

Minor Editorial comments:

Line 113, 137, Fig3A: Clec9a misspelt, should be changed to Clec9a.

Line 743: Typo for absolute

Line 326-329: This is an overstatement since the authors only evaluated PDL1+ cDC2 in the context of diet and XCR1+ cDC1 in the context of Abx.

Figure 4B: Typo in Y axis for Foxp3

Supplementary Figure 1c: Singlets spelling

Supplementary Figure 1d: Figure legend makes it unclear pie chart is summary data from the stomach or gut.

Reviewer #2 (Remarks to the Author):

This manuscript investigates tolerance induction in the intestine, with emphasis on the role of DC subsets, in the steady state and during inflammatory bowel disease models. The authors combined single cell RNA-sequencing and CyTOF cellular analysis to define signatures of DC subpopulations in distinct regions, including PD-L1, XCR1, IL-10, S100a4 and Sirpa, and link these signatures to Treg and Th17 cell differentiation and accumulation at the different regions of the intestine. The in-vivo finding that DC-specific ablation of PD-L1 or depletion of XCR1 DCs perpetuated the outcome of (5-FU)-induced mucositis and DSS-induced colitis delivers the functional relevance of these two DC subsets and their role in organ integrity under inflammatory conditions. Furthermore, loss of PD-L1+ DCs and XCR1+ DCs from the small intestine and colon after antigen-free diet or antibiotic treatment, respectively, indicated mechanistic relationships between food antigen and microbiome in the development and homeostasis of these DC subsets and further supports the notion that both Th17 and Tregs may be required for induction of complete tolerance in the intestine.

General remarks:

Overall, the manuscript is understandable to the expert, but several sections would benefit from a professional copyediting and restructuring/clarification. Most, but not all claims are supported by their data. Some supported claims in the beginning of the manuscript are not novel and there are a few inconsistencies that need to be resolved. The mechanistic elucidation stops at the identification of DC subsets responsible for the observed phenotype, but molecular mechanisms how DCs educate T cells in the intestine were not resolved. The experimental quality is very high mostly. The topic is of clinical relevance and might improve our understanding of establishment of tolerance to food and microbiome-derived antigens with respect to the role of intestinal DC subsets. The following major and minor points need to be addressed:

Major points:

Figure 1:

a. Line 146-149 and 120-122: Figure 1 shows a regionalized distribution of the PD-L1+ and XCR1+ DCs in the different compartments (i.e. duodenum and colon) of the intestine. However, the correlation between the distinct DC subsets and the tolerance to food antigens and microbiota antigens is an assumption. The authors need to rephrase their conclusions.

b. Although the t-SNE analysis Figure 1f shows a substantial increase in CD4 and CD8 T cells, the authors indicated that no differences were observed (line 107-108). This inconsistency needs to be resolved.

c. A main body of the data in Figure 1 dealing with the regional diversity of microbiota is a confirmation of what has been published previously (Ref #11 of the present manuscript) as well as in Martinez-Guryn K et al. Cell Host and Microbes (2019). These discoveries by others should be acknowledged. And since this part of the manuscript is not novel, it might as well be moved to the suppl. Figures.

d. The concept of region-specific distribution and abundance of CD11c, CD11b and different DC subsets as well as the correlation between the abundance of these APCs with the distribution of Tregs and Th17 cells in the different regions of the intestine has previously been shown by Denning TL et al; JI (2011). Also these discoveries by others should be acknowledged. The authors should state what is the added value of their data in Figure 1 and if there is none, it might be relocated to the suppl. material.

Figure 2:

Figure 2f is based entirely on assumptions. The authors show no experimental evidence that would justify placing this figure in the results section. This may be shown in the discussion.

Figure 3:

Line 146: the authors suggest the presence of two different cDC2s populations in different intestinal

compartments, which is contradictory to the conclusion drawn from results in Fig 3a-c. In lines 138-144 the authors suggest the accumulation of cDC1 cells in the large intestine and cDC2 in the small intestine. This inconsistency needs to be resolved.

Figure 4:

a. The spatial accumulation of Tregs and Th17 cells might be due to preferential recruitment of the T cells and not necessarily due to DC subtype-dependent T cell differentiation at the different regions of the intestine. The differentiation of the transferred naïve OT-II cell could take place in different inductive sites such the Peyer's patches, and then the T cells migrate to the intestinal lamina propria under the influence of IEC- or DC-derived chemokines. The authors have to show evidence that the differentiation of the Treg and T17 cells is induced by the different DC subsets at the different regions in situ. This is technically possible, for example by showing:

- interaction/co-localization of the OT-II cells with the XR1+ and PD-L1+ DCs at the different regions of the intestine by IF imaging techniques.
- Transfer of OT-II cells expressing Foxp3 and Rorc reporter genes and determine the kinetic of Treg and Th17 accumulation at the different regions of the intestine.

b. The finding in this Figure is consistent with the previously identified DC subsets with differential capacities to induce Tregs and Th17 cells at the LP of the small intestine [Timothy L Denning et al; Nature (2007) and Koji Atarashi et al; Nature (2008)], which should be acknowledged.

Figure 5:

a. It is unclear how the data in Figure 5 supports the conclusions the authors have drawn. their proposition that PD-L1 DCs are key for homeostasis maintenance and regulation of Treg differentiation in the small intestine, one wonders if PD-L1 ko as well as the CD11-Cre x PD-L1flox/flox mice show altered distribution and accumulation of Tregs in the duodenum? Do the mice show altered phenotype? This would be an important experimental test of their propositions.

b. The quality of the images from the human intestine sections is poor. It needs to be improved and further markers need to be included.

Figure 6: ok

Minor Points:

1. The flow in the result section should be improved. I would suggest putting the scRNA-seq data before the confirmation with the CyFOF analysis. The results move back and forth between the two of data Sets which is confusing.
2. Labelling (numbers) indicating main and supplemental figures are missing
3. Frequencies (2.0-3.9%) and (25.5-33.5%) in the duodenum and colon are only indicated for the CD11b but not for the CD11c (Figure 1g lower panel).
4. In Line 157: it is not clear what "data not shown" is referring to.

Reviewer #3 (Remarks to the Author):

The manuscript by Moreira et al addresses the interesting topic of the maintenance of intestinal immunological homeostasis by different populations of antigen presenting cells at different intestinal locations. The authors identify PD-L1+ and XCR1+ DCs as mediators of T cell homeostasis and intestinal inflammation, and that their function is influenced by microbial stimulation and antigenic exposure.

Major Comments:

1. Given the extensive nature of this manuscript and its findings, the introduction was very general and did not provide enough detail underlying the study design and aims. Little information is given on DC subsets and the maintenance of intestinal homeostasis, or the broader implications of the mechanisms by which this occurs. The introduction should also be expanded to include the major findings of the study.
2. The methods indicate that both male and female mice were used – given the importance of gender influences on immune function and microbial colonisation, this should be justified. Were genders matched for stimulators and responders in vitro stimulations?
3. Figure 1 parts f, g, and h need more explanation in the figure legend. Were the tSNE plots generated from a single mouse or from concatenated data? The calculations for Fig. 1j need explanation in the legend and Fig 1 k needs fully explaining in the text and legend, and axes need labelling.
4. Supp Fig 1a should show the data for CD24: given that changes in expression of this marker was one of the key findings of the study, it is surprising the data was not shown. Also, other key findings from this data should be fully discussed in the text referring to this figure, e.g. PD-L1 decrease. The legend for this figure is also incorrect, stating that part a should be tSNE plots from mass cytometry data, which it is not. The legend should also indicate how many mice these data are were derived from. Supp Fig 1b needs y-axis labels.
5. Figure 2d – the gating strategy for sorting DCs needs to be fully explained in the methods. How were the cells sorted, what were the recoveries and purities of the cells and what was the viability of cells from each location. Likewise, the purity and quality of RNA from sorted DCs should also be reported
6. Figure 2f indicates CD11c and CD11c expression by duodenal DCs, but not colonic DCs. In contrast, colonic DCs are indicated to express CD14, which is a prototypical marker of monocytes. Please provide confirmation that cells described as DC in the colon are actually DC, and are not either macrophages or monocytes or other myeloid-origin antigen presenting cells.
7. Figure 3a also appears to suggest that colonic DC do not express CD11c, although the colours are very difficult to interpret in this figure. Again, please provide conclusive evidence that the cells isolated from the colon were true DC and not another myeloid subset.
8. Figure 4a should indicate levels of T cell proliferation induced by the different DC subsets in the in vitro cultures, as cytokine production is dependent on levels of T cell activation and division. Ideally CFSE plots or similar should be shown here to indicate the extent of T cell division following stimulation.

Minor comments

1. All axes in supplementary figures (e.g. S1a and b) should be checked for correct labels
2. All supplementary figure legends should fully explain the experimental approach, including the numbers of animals used. If representative plots are shown, then indicate of how many individuals.
3. Supp Fig 2 and has two panels labelled "b" which should be corrected
4. The sentence at line 142 is incomplete

Reviewer #1

→ General comments:

This is a tremendous amount of work, much of it of high quality, that digs into the very complex question of how DC are influenced by their anatomical location, and whether this in turn impacts the “flavour” of the T cell response. Assessing these DC in different challenge situations (5-FU, DSS) and environmental changes (Abx, AA diet) was then used to ascertain how these DC adapt to various perturbations.

Due to the broad scope of the study, the paper is quite dense and sometime hard to follow with some internal inconsistencies in the data and problematic choices of mouse models. It would be far better if the manuscript were reduced in scope and authors shored up their findings on the first 5 figures.

→ Specific comments

“For example, if one were to simply focus on the anatomical differences in DC representation (an important finding in and of itself), there are some inconsistencies. Specifically, in Figure 4 authors state “These in vivo findings are consistent with our in vitro findings using DCs isolated from different regions of the gut to induced T cell differentiation”. However, this is not born out in the data: T cells stimulated in vitro with colonic DCs had an elevated Foxp3/Il17 ratio (**Figure. 4a-b**). This is in opposition to the in vivo observation that OTII-specific T cells adopt a Th17 phenotype in the colon (Fig. 4c-e)”

“In addition in Figure 5b-c the conclusion that there are greater numbers of XCR1+ DC in the colon per gram of tissue than PDL1+ DC does not compute with the representative flow cytometry plots displayed (eg, based on the frequencies of the representative plots in the colon, where PDL1+ DCs and XCR DCs make up 58% and 24% of CD11c-hi cells, respectively one would expect the opposite). Furthermore, the total number of DCs in the SI should be much larger than in the colon per g/tissue based on data in Fig. 1.”

° **Response part 1:** We agree with the reviewer that our results were inconsistent and to address this issue we performed an additional series of in vitro coculturing experiments to establish T cell differentiation at different DC:T cell ratios. Naive T cells were labeled with cell-trace dye to investigate T cell proliferation. We found Foxp3/ROR γ t differentiation at different DC:T cell dilutions (**new Figure 3b**). Consistent with our findings in vivo, we showed a higher Foxp3/ ROR γ t ratio in the upper SI versus colon. Similar findings by others were acknowledged and are highlighted in the text.

° **Response part 2:** We agree with the reviewer and to determine the number of DCs in different compartments of the gut, two additional experiments were performed. Discrepancies in cell number in Fig. 1 and Fig. 5 were likely related to the use of Itgax reporter mice and XCR1^{Venus} mice in different experiments. Graphs are now

representative of the same experiment using C57BL/6 mice only. Percentage and numbers are now consistent. (Please note previous **Figure 5** is now **Figure 4**).

“There could be many reasons for these inconsistencies. For example, while DCs are used in culture, macrophages make up a significant portion of APCs in the gut. These are not accounted for in the in vivo experiments. Moreover, physiological differences in the DC-T cell ratio (lower in the colon compared to the SI) might account for in vitro vs in vivo discrepancies. Does titrating down the DC:T cell ratio in vitro skew the T cells towards being more Th17 rather than regulatory? In these cultures, how was the ratio of DCs and T cells rationalized for the small intestines and the large intestines? In the mouse colon, the actual T cell to APC ratio is far greater than the mouse small intestines”

° **Response:** We agree with the reviewer that macrophages account for a significant portion of APCs in the gut and that this could have led to inconsistent results when compared to in vitro findings. However, it is difficult, if not impossible, to establish an in vitro system that perfectly resembles the in vivo environment, particularly the gut. Although macrophages can indeed work as APCs, DCs are the cells specialized in antigen presentation.

We also believe that the physiological differences in the DC:T cell ratio (lower in the colon compared to the SI) could account for in vitro vs in vivo discrepancies. To address this, we performed in vitro experiments using different DC:T cell titration. We found that the increase in DC:T ratio promoted greater differentiation of both Foxp3+ Tregs as well as ROR γ t+ T cells (**new Supplementary Fig 3b**). In the upper SI, higher DC:T cell ratios promoted increased Treg cell differentiation that was not observed in the colon (**new Figure 3b**). Very low or no T cell differentiation was found when a 1:1 or 1:5 DC:T cell ratio was applied. There was no difference between T cell proliferation when upper SI was compared to colon (**new Supplementary Fig 3c**), indicating that T cell proliferation is independent of DC:T cell ratio *in vitro*.

“In addition, where inflammatory insults are introduced (DSS, 5-FU) it is important for the authors to evaluate the immunosuppressive function of XCR1+ cDC1 and PDL1+ cDC2 across both the upper SI and the LI”

“As it stands, these analyses are incomplete. In general, the description of these mouse lines is also lacking. For example, the use of the CD11c-Cre is problematic – this will also delete PDL1 on CD11c+ macrophages. Authors need to fully characterize the expression of PDL1 on various APC subsets (including macrophages) and/or use a different more cDC-specific Cre animal”

° **Response part 1:** To address this, we performed two additional experiments to investigate local inflammation in XCR1 DTA mice using a DSS-colitis model (**Figure 5b**). We found that XCR1 DTA mice had increased colonic lamina propria infiltration of monocytes and neutrophils. Increased IL-17 production associated with increased ROR γ t+ T cell differentiation and reduction of Treg cells and T-bet expressing T cells were also found in the colonic lamina propria (**new Supplementary Figure 6c**). For PD-L1 conditional knockout mice, we also performed two additional experiments to

investigate local inflammation induced by 5-FU in mice lacking PD-L1 in DCs. In this case, greater monocyte and neutrophil infiltration was also found in the SI lamina propria of CD11c^{Cre}xPD-L1^{flox/flox} mice as compared to controls. Lamina propria T cells from CD11c^{Cre}xPD-L1^{flox/flox} mice also upregulated ROR γ t and increased IL-17 production. We also observed increased IFN- γ production in a T-bet independent fashion (**new Figure 5a; new Supplementary Figure 6c**). Of note, results found in the perturbation models using acute inflammation (DSS and 5-FU) are consistent with the *in vitro* experiments showing increased Th17 differentiation in naïve T cell coculture with colonic DCs sorted from XCR1 DTA mice, and increased Th17 and IFN- γ production in T cell coculture with upper SI DC sorted from PD-L1 conditional knockout (**new Supplementary Figure 5a,b**).

° **Response part 2:** We agree with reviewer and to address these concerns, we characterized the upper SI-LP from PD-L1 conditional knockout mice and results are shown in **new Supplementary Figure 6a**. PD-L1 MFI was reduced in intestinal DCs but not in intestinal macrophages. Additional data on characterization of PD-L1 expression in conditional knockout mice can be found in the work published by our collaborator Arlene Sharpe¹.

“In summary, the authors are urged to reconsider how to best “package” this work, perhaps developing the AA diet (which is very interesting!) and Abx perturbations for another study and focusing on shoring up the aforementioned issues in Figures 1-5. I believe that when tidied up, this more limited data set will be an important resource for the community and lays the groundwork for a separate study on diet and microbiome perturbations for the future”

° **Response:** As suggested, antibiotics and amino acid diet data were removed from the new version of the manuscript.

“Other comments that need to be addressed”

“**Figure 2a:** Interesting that Notch2 is upregulated in colonic DCs, as it's exclusively expressed by cDC2 (which are shown not to be the dominant DC population in this area). K. Murphy and colleagues show that these cDC2 are required for generating immune responses against *C. rodentium* in the SI. Please comment on this”

° **Response:** This is indeed an important point. It has been shown that Notch2 expression in cDC2 is transient and dependent on environmental factors^{2, 3}. We believe that in our experiments the colonic environment promoted the upregulation of Notch2 in cDC2. This may include for example, lymphotoxin beta that is known to be increased in colon patches and in ROR γ t + cells to modulate Notch2 expression in cDC2^{4, 5, 6}.

“**Figure 2d:** Why not also include levels of Sirpa (cDC2) in the RNA-seq data to assess changes across the different gut anatomical locations? This would provide a nice internal control validating the concept that cDC2 are more dominant in the SI vs the LI”

° **Response:** Although percentages of Sirpa is increased in upper SI (**Figure 2f**; duodenum 85% vs colon 69%) we do not find statistical differences in Sirpa gene expression in DC sorted from duodenum versus colon by RNA-seq analysis.

“**Figure 3a:** Not intuitive how to interpret these FlowSom plots. For example, CD209a (DC-SIGN) are previously shown to be down-regulated from the SI to the LI. But there are more red circles in the colon compared to the duodenum? Likewise, XCR1 in green is more visible in the duodenum than the colon. These data run counter to those in Figure 2D”

° **Response:** We agree that FlowSom plots are difficult to interpret, and we have elected to remove it from the revised manuscript.

“**Figure 4a:** It is appreciated that the cytokines measured here are often difficult to detect by Flow Cytometry, hence the use of ELISA. However, some of these cytokines are produced both by the T cells and the DCs themselves. I would recommend either showing the cytokine levels from DC co-cultured with CD4+ T cells that are not OVA-specific as a control or sorting the T cells post-culture and performing qPCR”

° **Response:** We have addressed the reviewer’s comment by sorting T cells post-culture for qPCR analysis. T cells were sorted from a coculture with DCs from upper SI and colon (1:20 ratio) for 7 days in the presence of OVA. Results were incorporated in **new Supplementary Fig 3a**. We found that *Ifng* mRNA was upregulated in T cells cocultured with colonic DCs and *Tnfa* mRNA was upregulated in T cells cocultured with upper SI DCs. *Il6* mRNA was not detected in post-cultured T cells and *Il10* mRNA expression was very low.

“**Figure 4b:** Please provide dot plots for Foxp3 and IL17 for completeness”

° **Response:** We provided dot plots for Foxp3 and IL-17 in **new Supplementary figure 3b**.

“**Supplementary Fig. 3:** Cytokines that were examined in both S3a and S3b should share the same Y axis for readers to make comparisons”

° **Response:** We now show in **new Supplementary Fig. 5a, b**, cytokines measured in supernatants from coculture of CD4 T cells and DCs sorted from PD-L1^{-/-} and XCR1^{-/-} mice and littermate controls. Because these experiments were performed on different days and with different mice and different intestinal compartments, it is not expected to have the same values for all cytokines measured and thus not practical to plot the same Y axis for all cytokines. Moreover, we did not compare cytokine levels between PD-L1^{-/-} and XCR1^{-/-} mice, but between transgenic mice and their respective littermate controls.

“**Figure 5J:** These data are not convincing. Quantification (#DC per um2 or some other metric) is required”

° **Response:** Immunofluorescence staining was performed using 3 colors only. Because we did not apply sufficient markers to distinguish DC from macrophage and B cells as well as tissue cDC1 and cDC2, we have elected to remove this data from the revised manuscript due to the difficulty to accurately quantify PD-L1 in DC.

“**Figure 6a-g:** Were XCR1-expressing cDC1s in either the upper SI or the LI perturbed by the AA-diet similar to PDL1-expressing DCs?”

° **Response:** As per the request of the reviewers, these AA data were removed from the revised manuscript.

“Figure 6h-n: Abx treatment definitely induces perturbations in DCs of the small intestine and should be similarly examined as the large intestine”

° **Response:** As per the request of the reviewers, these Abx data were removed from the revised manuscript.

“ **Figure 6 (general):** RNA-seq analysis should be performed looking at Sirpa+ cDC2 clusters and XCR1+ cDC1 clusters individually, rather than all DCs together. Brown et al. had previously established that Abx treatment reduces the abundance of Tbet-expressing cDC2, which are functionally described to be immunoregulatory, in both the SI and the LI. Since the authors describe XCR1+ cDC1 to be the main immunosuppressive DC subset in the LI, the authors should distinguish whether the Abx treatment is impacting XCR1+ cDC1 OR cDC2 or both to possess a pro-inflammatory gene signature. The DC-T cell co-culture assays should also be performed with Abx-treated cDC1 OR Abx-treated cDC2”

° **Response:** As per the request of the reviewers, these Abx data were removed from the revised manuscript.

→ **Minor Editorial comments:**

Line 113, 137, Fig3A: Clec9a misspelt, should be changed to Clec9a.

Line 743: Typo for absolute

Line 326-329: This is an overstatement since the authors only evaluated PDL1+ cDC2 in the context of diet and XCR1+ cDC1 in the context of Abx.

Figure 4B: Typo in Y axis for Foxp3

Supplementary Figure 1c: Singlets spelling

Supplementary Figure 1d: Figure legend makes it unclear pie chart is summary data from the stomach or gut.

° **Response:** We appreciate these detailed comments and have corrected these errors which are highlighted in the text.

Reviewer #2

→ **General comments:**

This manuscript investigates tolerance induction the intestine, with emphasis on the role of DC subsets, in the steady state and during inflammatory bowel disease models. The authors combined single cell RNA-sequencing and CyTOF cellular analysis to define signatures of DC subpopulations in distinct regions, including PD-L1, XCR1, IL-10, S100a4 and Sirpa, and link these signatures to Treg and Th17 cell differentiation and accumulation at the different regions of the intestine. The in-vivo finding that DC-specific ablation of PD-L1 or depletion of XCR1 DCs perpetuated the outcome of (5-FU)-induced mucositis and DSS-induced colitis delivers the functional relevance of these tow DC subsets and their role in organ integrity under inflammatory conditions. Furthermore, loss of PD-L1+ DCs and XCR1+ DCs from the small intestine and colon after antigen-free diet or antibiotic treatment, respectively, indicated mechanistic

relationships between food antigen and microbiome in the development and homeostasis of these DC subsets and further supports the notion that both Th17 and Tregs may be required for induction of complete tolerance in the intestine.

Overall, the manuscript is understandable to the expert, but several sections would benefit from a professional copyediting and restructuring/clarification. Most, but not all claims are supported by their data. Some supported claims in the beginning of the manuscript are not novel and there are a few inconsistencies that need to be resolved. The mechanistic elucidation stops at the identification of DC subsets responsible for the observed phenotype, but molecular mechanisms how DCs educate T cells in the intestine were not resolved. The experimental quality is very high mostly. The topic is of clinical relevance and might improve our understanding of establishment of tolerance to food and microbiome-derived antigens with respect to the role of intestinal DC subsets. The following major and minor points need to be addressed:

→ Specific Comments

“Major points”

“**Figure 1a:** Line 146-149 and 120-122: Figure 1 shows a regionalized distribution of the PD-L1⁺ and XCR1⁺ DCs in the different compartment (i.e duodenum and colon) of the intestine. However, the correlation between the distinct DCs subsets and the tolerance to food antigens and microbiota antigens is an assumption. The authors need to rephrase their conclusions”

° **Response:** This sentence has been modified and is highlighted in the text as follows: “Our results demonstrate that distinct cDC2s exist in different intestinal compartments and suggest that PDL1⁺ duodenal DCs play an important role in small intestinal tolerance and XCR1⁺ DC are important for tolerance in the large intestine.”

“**b.** Although the t-SNE analysis Figure 1f shows a substantial increase in CD4 and CD8 T cells, the authors indicated that no differences were observed (line 107-108). This inconsistency needs to be resolved”

° **Response:** As shown in **new Supplementary Figure 1b**, total CD4 and CD8 was unchanged but Treg cells were increased in the colonic LP compared to SI LP. We have modified the text to clarify this as follows: “No changes in **total** CD4⁺ and CD8⁺ T cells were observed”.

“**c.** A main body of the data in **Figure 1** dealing with the regional diversity of microbiota is a confirmation of what has been published previously (Ref #11 of the present manuscript) as well as in Martinez-Guryn K et al. Cell Host and Microbes (2019). These discoveries by others should be acknowledged. And since this part of the manuscript is not novel, it might as well be moved to the suppl. Figures”

° **Response:** We have moved microbiome data to **new Supplementary Figure 1** and acknowledged these references.

“**d.** The concept of region-specific distribution and abundance of CD11C, CD11b and different DC subsets as well the correlation between the abundance of the APC with the

distribution of Tregs and Th17 cells in the different regions of the intestine has previously been shown by Denning TL et al; JI (2011). Also, these discoveries by others should be acknowledged. The authors should state what is the added value of their data in **Figure 1** and if there is none, it might be relocated to the suppl. Material”

° **Response:** Denning TL et al; JI (2011) work is now acknowledged and highlighted in the text as follows: “Gut region-specific distribution and abundance of CD11c, CD11b and different DC subsets was also reported by Denning and colleagues⁷”. We used a panel containing 31 markers to identify the immune cell populations in the gut LP (and not only APC). Moreover, data validation using counting beads and tissue imaging also adds value to this figure and the manuscript and thus we have kept these findings as a main figure.

“**Figure 2:** Figure 2f is based entirely on assumptions. The authors show no experimental evidence that would justify placing this figure in the results section. This may be shown in the discussion.”

° **Response:** The reviewer is correct and we have removed the figure from the results section, placed it in supplementary data and refer to it in the discussion (**supplementary Fig 7a**).

“**Figure 3:** Line 146: the authors suggest the presence of two different cDC2s population in different intestinal compartments, which is contradictory to the conclusion drawn from results in Fig 3a-c. In lines 138-144 the authors suggest the accumulation of cDC1 cells in the large intestine and cDC2 in the small intestine. This inconsistency needs to be resolved.”

° **Response:** Please see our response to reviewer 1. cDC2 is dominant DC subtype in all compartments. Modified text was added to manuscript.

“Importantly, we found that PDL1⁺ DCs were enriched in the upper gut (84.1% in duodenum vs. 22% in the colon) whereas XCR1⁺ DCs were enriched in the colon (12% vs 30.3%, respectively) (**Fig. 2f**). cDC2s (Sirpa⁺) represented the most dominant DC subtype in all intestinal compartments, ranging from 85% of the DCs in the duodenum vs. 69% in the colon”

“**Figure 4 a.** The spatial accumulation of Tregs and Th17 cells might be due to preferential recruitment of the T cells and not necessarily due to DC subtype-dependent T cell differentiation at the different regions of the intestine. The differentiation of the transferred naïve OT-II cell could take place in different inductive sites such the Peyer's patches, and then the T cells migrate to the intestinal lamina propria under the influence of IEC- or DC-derived chemokines. The authors have to show evidence that the differentiation of the Treg and T17 cells is induced by the different DC subsets at the different regions in situ. This is technically possible, for example by showing:

- interaction/co-localization of the OT-II cells with the XCR1⁺ and PD-L1⁺ DCs at the different regions of the intestine by IF imaging techniques.
- Transfer of OT-II cells expressing Foxp3 and Rorc reporter genes and determine the kinetic of Treg and Th17 accumulation at the different regions of the intestine.”

Response: To address this we performed a new set of kinetic experiments in which we adoptively transferred naïve CD4 T cells to recipient mice to investigate T cell

differentiation in different compartments of the intestine as well as other lymphoid organs. For this, 2.5×10^6 naïve CD4 T cells from OT-II mice were transferred into CD45.1 congenic mice. Mice received an intrarectal injection of OVA in the following day (16h hours after T cell transfer) and OVA was given in the drinking water in the follow up period. Mice were sacrificed after 24h (1 day), 84h (3.5 days) and 180h (7.5 days) post cell transfer. Cells were harvested from spleen, Peyer's patches (PP), lower gut draining lymph nodes (iliac+caudal+cecum; ccLN), mesenteric lymph node (mLN), upper small intestine (upper SI) and colon. As shown in **new Supplementary Figure 4a**, the majority of adoptively transferred cells were found in the spleen after 24h and no cells were found in the colonic LP. Moreover, we did not detect any Foxp3 and ROR γ t staining at this time point in any of the adoptively transferred cells at all anatomical sites. Conversely, adoptively transferred cells were found in all sites investigated at day 3.5 with no differences in frequencies (**new Supplementary Figure 4b-d**). In this case, few Foxp3 or ROR γ t cells were detected in the spleen, mLN, ccLN and PP, although they were increased in small and large intestine. Numbers of transferred T cell expressing Foxp3 at day 7.5 were increased in both small and large intestines as compared to other lymphoid organs (**new Supplementary Fig 4d**). Interestingly, we found a greater number of adoptively transferred T cells in the PP at day 7.5 as compared to the upper SI, but with lower Foxp3 expression (**Supplementary Figure 4d**). Taken together, these data suggest that the gut is the preferential site for antigen-dependent T cell differentiation and therefore, intestinal DCs orchestrate T cell differentiation in situ when naïve T cells are adoptively transferred.

“b. The finding in this Figure is consistent with the previously identified DC subsets with differential capacities to induce Tregs and Th17 cells at the LP of the small intestine [Timothy L Denning et al; Nature (2007) and Koji Atarashi et al; Nature (2008)], which should be acknowledged.”

° Response: These works are now acknowledged in the text.

“Figure 5 a : It is unclear how the data in Figure 5 supports the conclusions the authors have drawn. their proposition that PD-L1 DCs are key for homeostasis maintenance and regulation of Treg differentiation in the small intestine, one wonders if PD-L1 ko as well as the CD11-Cre x PD-L1flox/flox mice show altered distribution and accumulation of Tregs in the duodenum? Do the mice show altered phenotype? This would be an important experimental test of their propositions.”

° Response: To address this, we investigated the distribution of Treg cells from naïve CD11-Cre x PD-L1flox/flox and littermate control mice and a new set of data was incorporated in the manuscript as **Supplementary figure 6a, b**. Cells were harvest from both upper SI LP and upper gut draining lymph nodes (pool of D1+D2+J) (Esterhazy, Nature, 2019)⁸. We did not find altered frequencies and/or numbers of Foxp3+ and ROR γ t+ T cells in the sites investigated (**new Supplementary Figure 6b**). It is important to note that PD-L1 expression is tightly regulated and is associated with cancer and inflammatory conditions^{1, 9, 10}. Consistent with this, we found that absence of PD-L1 in DCs led to increased cell infiltration and mucosal damage in 5-FU-induced mucositis (**Figure 5a**). PD-L1 may still be important for homeostatic mechanisms,

however, its role at steady-state conditions in specific-pathogen-free (SPF) mice may be less relevant.

Please note that we modified the text in line 219-221 from the original version of the manuscript as follows: “Thus, PD-L1⁺ and XCR1⁺ DCs play an important role in ~~maintaining homeostasis of~~ the small and large intestines, respectively, **under inflammatory conditions.**”

“b. The quality of the images from the human intestine sections is poor. It needs to be improved and further markers need to be included.

° **Response:** We agree with reviewer and because we did not identify reliable markers to distinguish cDC1 and cDC2 in human tissue, we removed this data from the revised the manuscript.

→ **Minor Editorial comments:**

1. The flow in the result section should be improved. I would suggest putting the scRNA-seq data before the confirmation with the CyFOF analysis. The results move back and forth between the two of data Sets which is confusing.

2. Labelling (numbers) indicating main and supplemental figures are missing

3. Frequencies (2.0-3.9%) and (25.5-33.5%) in the duodenum and colon are only indicated for the CD11b but not for the CD11c (Figure 1g lower panel).

Response: Percentages of frequencies for CD11c were located in the lower corner of t-SNE plots. We have aligned it with CD11b for better visualization.

4. In Line 157: it is not clear what “data not shown” is referring to.

° **Response:** We appreciate the reviewer’s detailed comments and have corrected these errors which are highlighted in the text.

Reviewer #3

→ **General comments:**

The manuscript by Moreira et al addresses the interesting topic of the maintenance of intestinal immunological homeostasis by different populations of antigen presenting cells at different intestinal locations. The authors identify PD-L1⁺ and XCR1⁺ DCs as mediators of T cell homeostasis and intestinal inflammation, and that their function is influenced by microbial stimulation and antigenic exposure

→ **Specific comments**

1) Given the extensive nature of this manuscript and its findings, the introduction was very general and did not provide enough detail underlying the study design and aims. Little information is given on DC subsets and the maintenance of intestinal homeostasis, or the broader implications of the mechanisms by which this occurs. The introduction should also be expanded to include the major findings of the study.

° **Response:** We agree with the reviewer and as also requested by the other two reviewers, to focus the manuscript we removed data on antibiotics and amino acid diet from the revised version. We have expanded the "Introduction" as suggested.

2) The methods indicate that both male and female mice were used – given the importance of gender influences on immune function and microbial colonization, this should be justified. Were genders matched for stimulators and responders in vitro stimulations?

° **Response:** All in vitro experiments and adoptive transfer experiments were performed using females only. For experiments in which we performed DSS-induced colitis and 5-FU-induced mucositis, we used both males and females considering equivalent N/gender per condition. We have added gender matching information to Animals-Methods (highlighted).

3) **Figure 1** parts f, g, and h need more explanation in the figure legend. Were the tSNE plots generated from a single mouse or from concatenated data? The calculations for Fig. 1j need explanation in the legend and Fig 1 k needs fully explaining in the text and legend, and axes need labelling.

° **Response:** Concatenated data were used, and this is now explained in the legend. IF and IHC quantifications were performed using batch image analysis using a custom Image J code that is now detailed in the Methods. Axes were labeled accordingly.

4) **Supp Fig 1a** should show the data for CD24: given that changes in expression of this marker was one of the key findings of the study, it is surprising the data was not shown. Also, other key findings from this data should be fully discussed in the text referring to this figure, e.g. PD-L1 decrease. The legend for this figure is also incorrect, stating that part a should be tSNE plots from mass cytometry data, which it is not. The legend should also indicate how many mice these data are were derived from. Supp Fig 1b needs y-axis labels.

° **Response:** CD24 data have been added to **Supplementary figure 1** as suggested, and also mentioned in the main text as follows: "Moreover, we found that 58.93% of live CD45+ cells in duodenum expressed CD24 vs 23.53% in the colon (**Supplementary Fig 2a**).” Other key findings are now also discussed in the text as follows: "We gated on CD45+ cells and found that markers associated with antigen presenting cells (APCs) including CD24, CD11b, CD11c and F4/80 as well as CD135, PD-L1, CD39, Sirpa, CCR7 and CD205 decreased from the upper to lower regions of the intestine with the highest numbers in the duodenum and lowest numbers in the colon". We apologize for the legend mistake. "t-SNE plots" were corrected to Histograms, and Y axis added.

5) **Figure 2d** – the gating strategy for sorting DCs needs to be fully explained in the methods. How were the cells sorted, what were the recoveries and purities of the cells and what was the viability of cells from each location? Likewise, the quality of RNA from sorted DCs should also be reported.

° **Response:** Cells were presorted using CD11c beads and sorted using flow cytometry. DCs were sorted as: Live, CD45+; T, B cells (CD90.2-, CD19-, B220-) and macrophages (F4/80-, CD64-) were dumped. CD11c^{Hi} cells were sorted as specified in

figure legend. Additional gating strategies were added as in **Supplementary Fig 2e-g**. Flow cytometry cell recovery and purity were higher than 97%. DCs were sorted from different intestinal compartments directly into lysis buffer, and RNA was extracted using PicoPure RNA Isolation Kit. High-recovery MiraCol™ Purification Columns were used. Contaminant ID and Purity ratios obtained by Nanodrop in one representative sample for each compartment showed good RNA quality and no DNA contaminants. Additional information related to RNA quality was added to Methods under RT-qPCR and is highlighted in the text. For RNA-seq samples, quality control was performed as described in the Methods and according to The Broad Institute quality control standard procedures.

6) Figure 2f indicates CD11c and CD11c expression by duodenal DCs, but not colonic DCs. In contrast, colonic DCs are indicated to express CD14, which is a prototypical marker of monocytes. Please provide confirmation that cells described as DC in the colon are actually DCs and are not either macrophages or monocytes or other myeloid-origin antigen presenting cells.

° **Response:** In response to Reviewer 2, Figure 2f was removed from the main figures and is now in supplementary data. CD14 was represented in “smaller letters” meaning low expression. CD14 expression may be found in DCs because of its myeloid origin. For RNA-seq analysis, cells were sorted as specified above and macrophages were dumped using F4/80, CD64 and CD11c^{high} staining (**new Supplementary Figure 7a**).

7) **Figure 3a** also appears to suggest that colonic DC do not express CD11c, although the colours are very difficult to interpret in this figure. Again, please provide conclusive evidence that the cells isolated from the colon were true DC and not another myeloid subset.

° **Response:** As suggested by Reviewer 1, considering the DC heterogeneity and the difficulty to identify colors and size in spanning tree branches, we have removed this analysis from the revised manuscript. Again, macrophages were dumped using F4/80, CD64 and CD11c^{high} staining (**new Supplementary Fig 2e-g**).

8) **Figure 4a** should indicate levels of T cell proliferation induced by the different DC subsets in the in vitro cultures, as cytokine production is dependent on levels of T cell activation and division. Ideally CFSE plots or similar should be shown here to indicate the extent of T cell division following stimulation.

° **Response:** To address this question, we performed an additional series of in vitro coculturing experiments to establish T cell differentiation at different DC:T cell ratios. Naive T cells were labeled with cell-trace dye to address T cell proliferation. We found Foxp3/ RORγt differentiation at different DC:Tcell dilutions. Consistent with our findings in vivo, we found higher Foxp3/RORγt ratios in the upper SI versus colon. In addition, we found no differences in T cell proliferation between upper SI and colon assessed by cell-trace staining (**new Supplementary Figure 3c**).

→ **Minor Editorial comments:**

1. All axes in supplementary figures (e.g. S1a and b) should be checked for correct labels

2. All supplementary figure legends should fully explain the experimental approach, including the numbers of animals used. If representative plots are shown, then indicate of how many individuals.
3. Supp Fig 2 and has two panels labelled “b” which should be corrected
4. The sentence at line 142 is incomplete

° **Response:** We appreciate the reviewer’s detailed comments and have corrected these errors which are highlighted in the text.

REFERENCES

1. Sage, P.T. *et al.* Dendritic Cell PD-L1 Limits Autoimmunity and Follicular T Cell Differentiation and Function. *J Immunol* **200**, 2592-2602 (2018).
2. Lewis, K.L. *et al.* Notch2 receptor signaling controls functional differentiation of dendritic cells in the spleen and intestine. *Immunity* **35**, 780-791 (2011).
3. Satpathy, A.T. *et al.* Notch2-dependent classical dendritic cells orchestrate intestinal immunity to attaching-and-effacing bacterial pathogens. *Nat Immunol* **14**, 937-948 (2013).
4. Macho-Fernandez, E. *et al.* Lymphotoxin beta receptor signaling limits mucosal damage through driving IL-23 production by epithelial cells. *Mucosal Immunol* **8**, 403-413 (2015).
5. Wang, Y. *et al.* Lymphotoxin beta receptor signaling in intestinal epithelial cells orchestrates innate immune responses against mucosal bacterial infection. *Immunity* **32**, 403-413 (2010).
6. Taylor, R.T., Lugering, A., Newell, K.A. & Williams, I.R. Intestinal cryptopatch formation in mice requires lymphotoxin alpha and the lymphotoxin beta receptor. *J Immunol* **173**, 7183-7189 (2004).
7. Denning, T.L. *et al.* Functional specializations of intestinal dendritic cell and macrophage subsets that control Th17 and regulatory T cell responses are dependent on the T cell/APC ratio, source of mouse strain, and regional localization. *J Immunol* **187**, 733-747 (2011).
8. Esterhazy, D. *et al.* Compartmentalized gut lymph node drainage dictates adaptive immune responses. *Nature* **569**, 126-130 (2019).
9. Ganesan, A. *et al.* Comprehensive in vitro characterization of PD-L1 small molecule inhibitors. *Sci Rep* **9**, 12392 (2019).
10. Francisco, L.M. *et al.* PD-L1 regulates the development, maintenance, and function of induced regulatory T cells. *J Exp Med* **206**, 3015-3029 (2009).

REVIEWERS' COMMENTS

Reviewer #1 (Remarks to the Author):

The authors have added considerable new data and the manuscript is much more focused. Thank you. There are a few remaining issues that need to be resolved to ensure that the conclusions are in line with the data.

1. The new data in Fig. 5 support the concept that PDL1+ DC are important for resolving 5-FU mediated inflammation, and XCR1+ DC are important for resolving DSS mediated inflammation. However, these data do not preclude the possibility that PDL1+ DC are also involved in limiting DSS inflammation and XCR1+ DC are also involved in limiting 5-FU mediated inflammation as the converse experiments were not performed. After all, these 2 DC subsets are found in both locations. For that matter, other myeloid cells, if removed, may likewise have an impact in these models. Authors need to modify their conclusions accordingly.

2. Fig. 4 b/c is presented in a very confusing way. Authors need to plot the RATIO of PD-L1+/XCR1+ DC in both the small intestine and colon.

Addressing these concerns need not require new data but rather a shoring up of the text.

Reviewer #2 (Remarks to the Author):

The authors have adequately addressed all my questions and concerns. they have performed all required experiments and they supported the claims of the manuscript.

Reviewer #3 (Remarks to the Author):

I thank the authors for their detailed responses to my comments, and the additional experimental work to address certain issues. While a number of my concerns were shared and addressed in responses to the other reviewers, I am happy that one of my key concerns regarding the correct identification and isolation of DC was addressed, as this is an aspect core to the manuscript. I do appreciate the technical challenges of sorting DC from mouse mucosal surfaces, and also the potential for contamination of "non-DC" myeloid cells in these preps that can skew the data, especially if CD11c is used as a primary identifier. However, I am satisfied the authors have addressed this, and while not a barrier to publication, visual confirmation of the purity of the sorted DCs would have added further strength to this aspect of the manuscript. Nevertheless, this is a very comprehensive and important study, and I would be happy to see it published in its revised form.

REVIEWERS' COMMENTS

Reviewer #1

→ General comments:

The authors have added considerable new data and the manuscript is much more focused. Thank you. There are a few remaining issues that need to be resolved to ensure that the conclusions are in line with the data.

→ Specific comments

1. The new data in Fig. 5 support the concept that PDL1+ DC are important for resolving 5-FU mediated inflammation, and XCR1+ DC are important for resolving DSS mediated inflammation. However, these data do not preclude the possibility that PDL1+ DC are also involved in limiting DSS inflammation and XCR1+ DC are also involved in limiting 5-FU mediated inflammation as the converse experiments were not performed. After all, these 2 DC subsets are found in both locations. For that matter, other myeloid cells, if removed, may likewise have an impact in these models. Authors need to modify their conclusions accordingly.

° **Response:** We agree with reviewer comment and have added to the manuscript the following sentence: "Of note, because PD-L1+ and XCR1+ DC are found in all intestinal compartments, we cannot rule out they limit inflammation beyond their site of enrichment".

2. Fig. 4 b/c is presented in a very confusing way. Authors need to plot the RATIO of PD-L1+/XCR1+ DC in both the small intestine and colon.

° **Response:** As stated by the editor, we did not address this comment since ratios can be calculated from numbers presented.

Final comment: Addressing these concerns need not require new data but rather a shoring up of the text.

Reviewer #2

→ General comments: The authors have adequately addressed all my questions and concerns. they have performed all required experiments and they supported the claims of the manuscript.

° **Response:** No additional requests by reviewer 2.

Reviewer #3

→ General comments: I thank the authors for their detailed responses to my comments, and the additional experimental work to address certain issues. While a number of my concerns were shared and addressed in responses to the other reviewers, I am happy that one of my key concerns regarding the correct identification and isolation of DC was addressed, as this is an aspect core to the manuscript. I do appreciate the technical challenges of sorting DC from mouse mucosal surfaces, and also the potential for

contamination of "non-DC" myeloid cells in these preps that can skew the data, especially if CD11c is used as a primary identifier. However, I am satisfied the authors have addressed this, and while not a barrier to publication, visual confirmation of the purity of the sorted DCs would have added further strength to this aspect of the manuscript. Nevertheless, this is a very comprehensive and important study, and I would be happy to see it published in its revised form

° **Response:** No additional requests by reviewer 3.

Supplementary figures showing gating strategies confirming dumping of several potential contamination of "non-DC" (eg: F4/80, CD64, B220, CD19, CD3) is provided in the manuscript. Sorting efficiency of DC was greater than 95% purity in all experiments.